# Simultaneous analytical method for 296 pesticide multiresidues in root and rhizome based herbal medicines with GC-MS/MS

**Seung-Hyun Yang**[1,2°]**, Yongho Shin**[3°]**, Hoon Choi**[1]*

**1** Department of Life & Environmental Sciences, College of Agriculture and Food Sciences, Wonkwang University, Iksan, Republic of Korea, **2** Department of Healthcare Advanced Chemical Research Institute, Environmental Toxicology & Chemistry Center, Hwasun-gun, Republic of Korea, **3** Department of Applied Biology, College of Natural Resources and Life Science, Dong-A University, Busan, Republic of Korea

☯ These authors contributed equally to this work.

* hchoi0314@wku.ac.kr

**Data Availability Statement:** All relevant data are within the manuscript and its Supporting Information files.

## Abstract

A method for the simultaneous analysis of pesticide multiresidues in three root/rhizome-based herbal medicines (*Cnidium officinale*, *Rehmannia glutinosa*, and *Paeonia lactiflora*) was developed with GC-MS/MS. To determine the concentrations of pesticide residues, 5 g of dried samples were saturated with distilled water, extracted with 10 mL of 0.1% formic acid in acetonitrile/ethyl acetate (7:3, v/v), and then partitioned using magnesium sulfate and sodium chloride. The organic layer was purified with Oasis PRiME HLB plus light, followed by a cleanup with dispersive solid-phase extraction containing alumina. The sample was then injected into GC-MS/MS (2 μL) using a pulsed injection mode at 15 psi and analyzed using multiple reaction monitoring (MRM) modes. The limit of quantitation for the 296 target pesticides was within 0.002–0.05 mg/kg. Among them, 77.7–88.5% showed recoveries between 70% and 120% with relative standard deviations (RSDs) ≤20% at fortified levels of 0.01, and 0.05 mg/kg. The analytical method was successfully applied to real herbal samples obtained from commercial markets, and 10 pesticides were quantitatively determined from these samples.

## Introduction

Herbal medicines are generally defined as the parts of plants or their complex mixtures having biologically active ingredients [1]. For over 3,000 years, they have been used to treat a wide range of symptoms and ailments, including colds, headaches, menstrual problems, asthma, and other immune problems, liver disease, and various cancers [2]. Originating in Asia, herbal medicines are also called Chinese herbal medicine (CHM) in China, Kampo-yaku in Japan, and Hanyak in Korea, and have become a popular alternative to synthetic pharmaceutical drugs in Western countries [3].

Thousands of herbal medicines have been identified, and it is known that 200–600 CHM are commonly used [4]. Among them, *Cnidium officinale*, *Rehmannia glutinosa*, and *Paeonia*

**Funding:** This study was supported by a grant (21172MFDS149) from the Ministry of Food and Drug Safety for the draft submission, and there was no additional external funding received for this study during revision period. The funders had no role in study design, data collection and analysis, decision to publish, or preparation of the manuscript.

**Competing interests:** The authors have declared that no competing interests exist.

*lactiflora* are popular in the Republic of Korea, where they are ranked in the top 10 based on criteria of cultivation area, production, and consumption [5, 6]. Generally, dried roots or rhizomes of these herbal medicines are collected and extracted using a water decoction. This process is different from that of modern medicines, in which only certain active ingredients are purified and prescribed. As a result, any impurities unintentionally introduced during cultivation, such as pesticides and heavy metals, can be co-extracted during the preparation of these herbal medicines.

Many edible herbal plants, classified as special crops and consumed in specific cases, do not have established pesticide maximum residue limits (MRLs) like common crops in most countries. According to the pharmacopoeia of Korea, it is recommended to evaluate the risk of pesticides based on the daily dose of herbal medicines and the acceptable daily intake (ADI) of pesticide residues, but only a few compounds are regulated under an official rule. In China, MRLs have been established for some pesticides under the 'Herbs' category, most of which are classified as Ginseng MRLs [7]. Other pesticides have not been classified in detail. However, it has been reported that various pesticides have been detected in herbal medicines in Korea and Asian countries [8–11]. Therefore, it is crucial to continuously monitor pesticide residue levels in herbal medicines for risk management.

To detect pesticide multiresidues in root/rhizome-based herbal medicines, it is necessary to develop an analytical method tailored specifically to their unique characteristics, which differ from those of general crops. Roots and rhizomes of herbal medicines possess complex matrices abundant in biologically active ingredients, secondary metabolites, and various phytochemicals [12]. However, during sample preparation, the co-extraction of these phytochemicals with pesticides can hinder the precise determination of pesticide residues. Therefore, these compounds should be removed via proper sample preparation.

Acetonitrile (ACN) is the major extraction solvent in the traditional Quick, Easy, Cheap, Effective, Rugged, and Safe (QuEChERS) procedures for common crops [13]. Various versions of QuEChERS procedures have been adopted to extract pesticides from root/rhizome-based herbal medicines [8, 14–17]. The polar chemical properties of ACN strongly exclude unwanted interferences, whereas the extraction efficiency for multiresidues can be reduced in samples with complex matrices [18, 19]. Another way to extract pesticides from herbal medicines is to use ethyl acetate (EA) [20]. The extraction efficiency of EA is often higher than that of ACN, but it can co-extract a significant amount of nonpolar interferences, such as lipids and epicuticular wax material [21]. ACN and EA are miscible, and by adjusting their ratio, a desired polar solvent mixture can be obtained [22]. Therefore, a combination of ACN and EA is expected to reduce their weaknesses and enhance their strengths. In addition, the selection of the optimal sorbents for dispersive-solid phase extraction (d-SPE) or a combination of two cleanup procedures effectively remove sample matrices without a severe loss of multiresidues.

In this study, an analytical method for the determination of 296 multi-residual pesticides in *C. officinale*, *R. glutinosa*, and *P. lactiflora* was developed using gas chromatography-tandem mass spectrometry (GC-MS/MS). Most of the targeted pesticides are regulated as MRLs in food in both the Republic of Korea and China. To optimize the sample preparation method, various extraction solvents and cleanup sorbents were compared. Using the established method, a quantitative analysis for the target pesticides was conducted using herbal samples obtained from commercial markets. This was the first trial to simultaneously analyze nearly 300 pesticides in *C. officinale*, *R. glutinosa*, and *P. lactiflora*. It is also worthwhile for developing a novel preparation procedure, which includes an acidified ACN/EA mixture during extraction and a combination of Oasis PRiME HLB plus light and alumina d-SPE for sample cleanup.

## Materials and methods

### Reagents, materials, and samples

Stock solutions of the target pesticides (analytical grade) were purchased from Kemidas Standard (Gunpo, Republic of Korea). Acetonitrile (ACN; HPLC grade) was obtained from J. T. Baker (Centre Valley, PA, USA). HPLC grade ethyl acetate (EA) and magnesium sulfate ($MgSO_4$; ≥99%) were purchased from Daejung Chemicals & Metals (Siheung, Republic of Korea). Formic acid (≥99%), acetic acid (≥99%), and sodium chloride (NaCl; ≥97%) were obtained from Junsei Chemicals (Tokyo, Japan). Ceramic homogenizers for 50-mL tubes and various dispersive-solid phase extraction (d-SPE) tubes, including Part No. 5982–4921 (25 mg C18 and 150 mg $MgSO_4$), 5982–5021 (25 mg primary secondary amine; PSA and 150 mg $MgSO_4$), 5982–5121 (25 mg PSA, 25 mg C18, and 150 mg $MgSO_4$), 5982–5122 (50 mg PSA, 50 mg C18, and 150 mg $MgSO_4$), 5982–5221 (25 mg PSA, 2.5 mg graphitized carbon black; GCB, and 150 mg $MgSO_4$), and 5982–5321 (25 mg PSA, 7.5 mg GCB, and 150 mg $MgSO_4$), were purchased from Agilent technology (Santa Clara, CA, USA). Alumina (≥99%) was purchased from Sigma-Aldrich (St. Louis, MO, USA). The Oasis PRiME HLB cartridge plus light (100 mg) was obtained from Waters Corporation (Milford, MA, USA). Deionized water (18.2 MΩ/cm) was prepared in-house using a Direct-Q3 UV (Darmstadt, Germany).

To develop and validate the analytical method, three herbal medicines (*C. officinale*, *R. glutinosa*, and *P. lactiflora*.) were obtained from Humanherb (Daegu, Republic of Korea). Pesticides were confirmed to be absent in these samples using three different versions of the QuEChERS method [13, 21, 23].

Real samples were obtained from various commercial markets, where two types of origins (Republic of Korea and China) were available. Among the 47 collected samples, the numbers of *C. officinale*, *R. glutinosa*, and *P. lactiflora* from Korea were 14, 8, and 10, respectively, and those from China were 5, 7, and 3, respectively. All samples were chopped properly, homogenized using a mixer, and stored at –20˚C until preparation.

### Preparation of the matrix-matched standard solutions

The pesticide stock solutions were mixed such that the concentration of each pesticide was 2500 ng/mL. The solution was diluted with ACN to obtain working solution concentrations of 500, 100, 50, 25, 10, 5, 2.5, and 1 ng/mL. For GC-MS/MS analysis, signal enhancement or suppression of the target pesticides by the sample matrices was corrected using matrix-matched standard calibration. Each working solution was mixed in a 1:1 ratio (v/v) with the extract solution obtained from the pesticide-free control samples, and the concentrations of the matrix-matched standards were 0.5, 1.25, 2.5, 5, 12.5, 25, and 50 ng/mL. All working solutions and matrix-matched standard solutions were stored at –20˚C until analysis.

### GC-MS/MS parameters

The GC-MS/MS conditions were modified from the instrumental methodology of Park et al. (2022) [24]. Pesticide multiresidues were analyzed on an Agilent 7890 B gas chromatograph system coupled with an Agilent 7000C triple quadrupole mass spectrometer (Agilent Technologies, Santa Clara, CA, USA). Chromatographic separation was performed using a DB-5 MS UI column (30 m L. × 0.25 mm I.D., 0.25 μm film thickness, Agilent technology). Helium (≥ 99.999%) was selected as the carrier gas, and its constant flow was 1.5 mL/min. The injection port temperature was 280˚C and the injection mode was splitless. The oven temperature program was initiated at 60˚C (held for 3 min), ramped to 180˚C at 20˚C/min (held for 3 min), increased to 260˚C at 15˚C/min (held for 3 min), and then increased to 300˚C at 10˚C/

min (held for 6 min). The total analysis time was 32.0 min. The pulsed injection of GC was tested at 5, 15, 25, 35, and 45 psi, and the signal intensities of the target pesticides at 25 ng/mL were compared with those in the unpulsed condition.

The mass spectrometer system was operated in the electron ionization (EI) mode at 70 eV. The ion source and transfer line temperatures were 230 and 280°C, respectively. Nitrogen ($\geq$99.999%) was used as the CID gas. The detector voltage was set at 1.4 kV. The qualitative/quantitative data were processed using Mass Hunter Workstation software Quantitative Analysis for QQQ (Version B.08.00). The MRM mode of GC-MS/MS was used to analyze the target pesticides. The detailed MRM transitions, collision energy (CE), and retention times of the target compounds are listed in S1 Table.

## Comparison of sample extractions

Samples (5 g) were extracted with four types of solvents (10 mL): ACN, ACN/EA (7:3, v/v), ACN/EA (3:7, v/v), and EA. Each extract was partitioned by adding 4 g of $MgSO_4$ and 1 g NaCl, and without cleanup steps. Each ACN layer was dried with a nitrogen stream at 40°C, and the dry matter from the four types of solvents was compared. This process was repeated nine times ($n$ = 9) for each type of solvent. Additionally, each sample underwent a recovery test. A 5 g sample was treated with 100 μL pesticide working solution to give 0.05 mg/kg concentration for target pesticides, and extracted following the corresponding procedures. The recovery rate (%) was determined as the ratio of the signal (area) of the target compound in the recovery sample to that in the matrix-matched standards. This study was also repeated three times ($n$ = 3) for each type of samples.

For the evaluation of acid efficiency during extraction, 0.1, 0.4, and 1% formic acid or acetic acid were added to ACN/EA (7:3, v/v), and the sample was extracted in the corresponding solvents, and then partitioned with 4 g of $MgSO_4$ and 1 g NaCl. Recoveries in each condition were compared at 0.05 mg/kg ($n$ = 3). In the overall extraction studies, we compared the extraction patterns across three types of herbal medicines: *C. officinale*, *R. glutinosa*, and *P. lactiflora*.

## Comparison of sample cleanup

The extracted samples from the optimized extraction step were purified using various types of d-SPE sorbents containing 150 mg $MgSO_4$: (1) 25 mg PSA, (2) 25 mg C18, (3) 25 mg PSA and 25 mg C18, (4) 50 mg PSA and 50 mg C18, (5) 25 mg PSA and 2.5 mg GCB, (6) 50 mg PSA and 7.5 mg GCB, and (7) 25 mg alumina, and cleanup with (8) Oasis PRiME HLB plus light. Recoveries under each condition were compared at 0.05 mg/kg. The matrix effect of each analyte was assessed by comparing the calibration slope from the solvent-based standard solutions (*a*) and that from the corresponding matrix-matched standard solutions (*b*). The matrix effect value (%) was calculated using the equation *(ME, %) = (b/a − 1) × 100*. In addition, a dual purification was conducted on the extract obtained from the No. (8) procedure by further implementing the No. (2) or (7) purification methods. The purification efficiency of each preparation method was compared in three types of herbal medicines, taking into account both recovery rates and matrix effects.

## Established sample preparation method

The homogenized sample (5 g) in a 50-mL centrifuge tube was saturated with 10 mL of distilled water for 30 min to ensure sufficient soaking. The samples were then extracted with 10 mL of ACN/EA solvent (7:3, v/v) and shaken for 3 min at 1,300 rpm using a Geno/Grinder (Spex SamplePrep, Metuchen, NJ, USA). The extract was added with 4 g $MgSO_4$ and 1 g NaCl,

shaken for 1 min at 1,300 rpm, and centrifuged for 5 min at 4,000 rpm using a centrifuge Combi-408 (Hanil Science Inc., Gimpo, Republic of Korea). In this step, the water residue was isolated through liquid-liquid partitioning, which allowed for the capture of polar co-extracts in the aqueous layer. According to the manufacturer's instructions and methods outlined in a previous paper [25], the organic supernatant (2 mL) was loaded into a syringe connected to the Oasis PRiME HLB plus light and passed through the cartridge. One milliliter of the eluate was transferred to a microcentrifuge tube containing 150 mg of $MgSO_4$ and 25 mg of alumina, vortexed for 1 min, and then centrifuged at 13,000 rpm for 5 min using a microcentrifuge M15R (Hanil Science Inc., Gimpo, Republic of Korea). The upper layer (500 μL) was mixed with ACN (500 μL) for matrix matching. The sample was equivalent to 0.25 g per 1 mL in the final extract. The final extract solution was injected into the GC-MS/MS system (2 μL).

## Method validation

The optimized analytical method was validated using the limit of quantitation (LOQ), linearity of calibration curve, and recovery. The LOQ was evaluated as the lowest concentration satisfying a signal-to-noise ratio (S/N) above 10. The linearity of the calibration curve (2–50 ng/mL) for each pesticide was evaluated using the correlation coefficient ($r^2$). Recovery ($n = 3$) was studied by spiking 100 μL of two levels of standard solutions (500 and 2500 ng/mL) into 5 g of the control sample, followed by preparation as the final established procedure, and then comparing the analyte area with that of matrix-matched standard calibration. Fortification levels were 0.01 and 0.05 mg/kg sample, which are equivalent to 2.5 and 12.5 ng analyte per mL extract solution, respectively.

# Results and discussion

## Comparison of the sensitivities of target analytes in various pulsed injections

Pulsed-splitless injection of GC can improve the shape and sensitivity of the target peaks by establishing the optimal inlet pressure during sample injection. In many studies analyzing pesticides using GC systems, the pulsed injection mode was used [26–28]. Ling et al. reported that the peak heights of methamidophos, acephate, and omethoate improved using pulsed splitless injection at 30 psi [27]. Godula et al. (1999) recommended not using a pulsed pressure exceeding 60 psi to obtain good responses for all analytes, including early eluting pesticides [28].

A comparison of the peak areas of the 296 target analytes at 5, 15, 25, 35, and 45 psi in the unpulsed mode revealed a characteristic intensity pattern for four retention time ($t_R$) segments (8–14, 14–16.2, 16.2–18, and 18–25 min), as shown in S1 Fig. At the 5 psi pressure pulse, the average relative intensities were less than 74% of those observed in the unpulsed injection (100%). At 25, 35, and 45 psi pressure pulses, the average relative intensities were inferior (58–96%) to those at the unpulsed injection in the $t_R$ range of 8–16.2 min, but the magnitude of the relative intensities improved at longer retention times, showing more than 109% at 16.2–25 min. Furthermore, the differences in the average relative intensities for each $t_R$ increased as the pulse pressure increased. Therefore, the tested pulsed pressures are not suitable as multiresidue instrument conditions since they do not improve the intensities of the analytes in the overall $t_R$. However, when the pulse pressure was set to 15 psi, the average relative intensities (102–112%) increased compared to those in the unpulsed condition in the overall $t_R$ ranges. Therefore, a pulse pressure of 15 psi was applied in the established analytical method.

## Optimization of sample extraction

Extraction solvents were selected based on their polarity and ability to extract a wide range of pesticide residues with varying polarities. Both ACN and EA are organic solvents with high solubility for pesticides. They are popular and representative solvents for extracting pesticide multiresidues [29]. In this study, ACN, EA, and their mixtures were tested as extraction solvents, and their extraction efficiencies were compared. The first evaluation involved weighing the dry matter of extracts from the control samples (Fig 1). All extracts from *C. officinale*, *R. glutinosa*, and *P. lactiflora* showed that the weights of their dry matter increased as the ratio of EA in the solvent increased. The ideal solvent should aim to extract as many target pesticides as possible with high efficiency while excluding unnecessary matrices in the samples. When the solvent extracts too many non-volatile materials (salts, sugars, proteins, etc.), chromatographic problems can occur for some analytes, and severe contamination in ion sources and quadrupoles of mass spectrometry can occur [30]. Our study indicated that EA is more likely to co-extract unnecessary interference, compared to ACN. Anastassiades et al. (2003) also pointed out that EA co-extracts a large amount of unnecessary nonpolar matrices, such as lipids and epicuticular wax material [21]. Therefore, extraction solvents with a low EA ratio can avoid frequent contamination of analytical instruments and the consequent replacement of consumables.

The second evaluation was the determination of the recovery rates of the target analytes. According to Shin et al.'s (2020) method [18], pesticides showing a difference between the maximum and minimum recovery rates greater than 25% in the four extraction solvents were verified (Fig 2). In the *C. officinale* sample, 14 compounds were sorted, 13 of which showed excellent recoveries greater than 70% in the ACN/EA mixtures with 7:3 (v/v) and 3:7 ratios, respectively (Fig 2A). Profluralin in the 7:3 mixture and thiometon in the 3:7 mixture were the only compounds that did not meet the criteria. However, in ACN and EA extraction, only 7

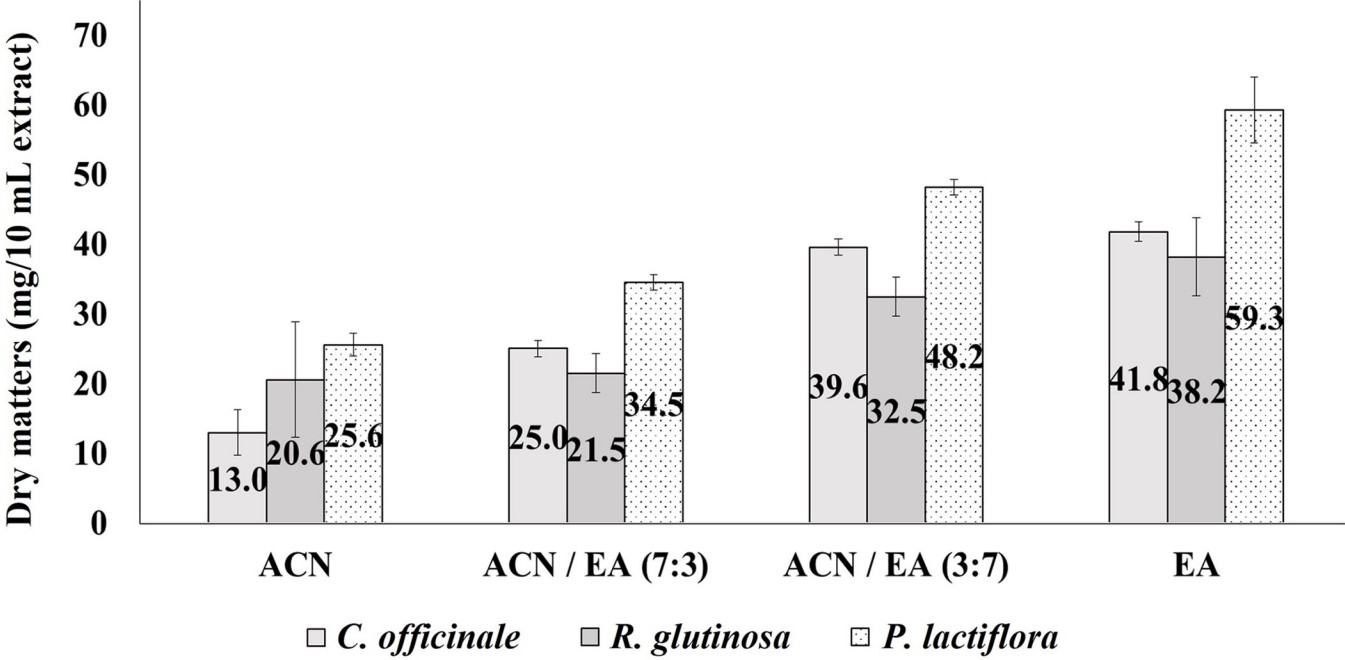

**Fig 1. Dry matters (*n* = 9) of *Cnidium officinale*, *Rehmannia glutinosa*, and *Paeonia lactiflora* from 10 mL extracts using acetonitrile (ACN), ACN/ethyl acetate (EA) (7:3, v/v), ACN/EA (3:7, v/v), and EA as extraction solvents.**

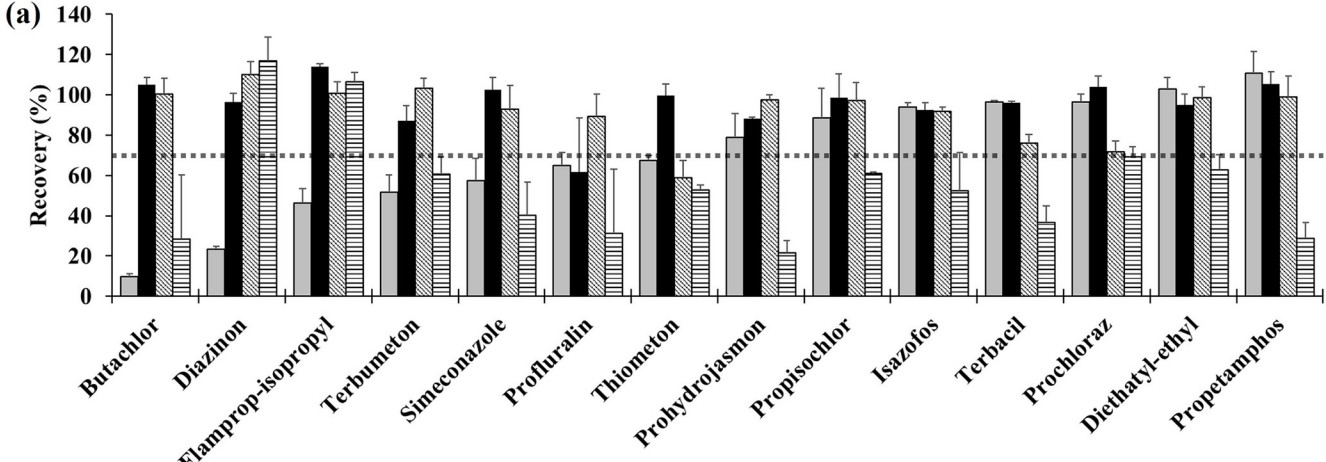

Pesticide in *Cnidium officinale*

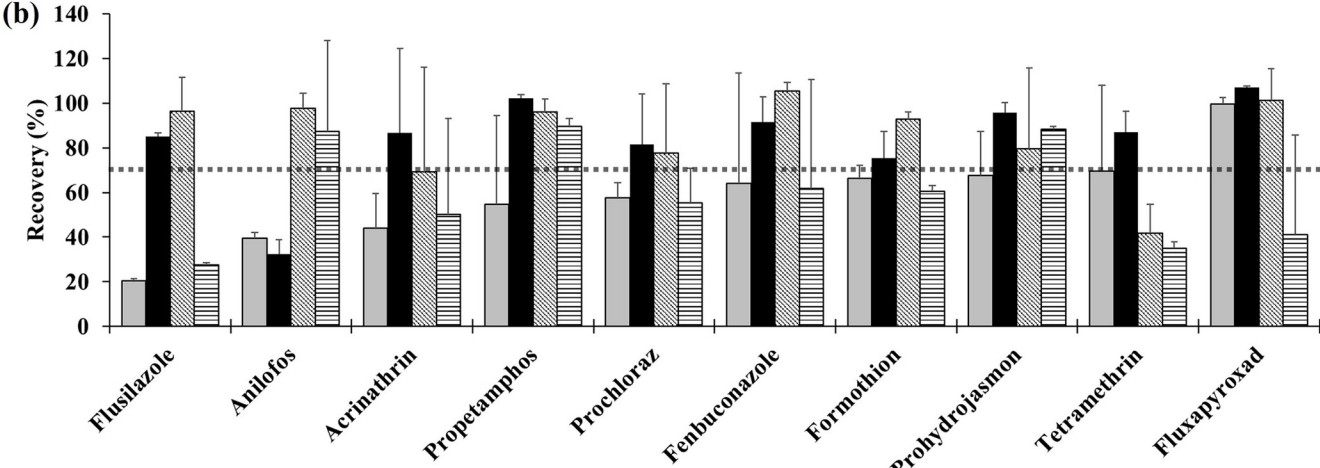

Pesticide in *Rehmannia glutinosa*

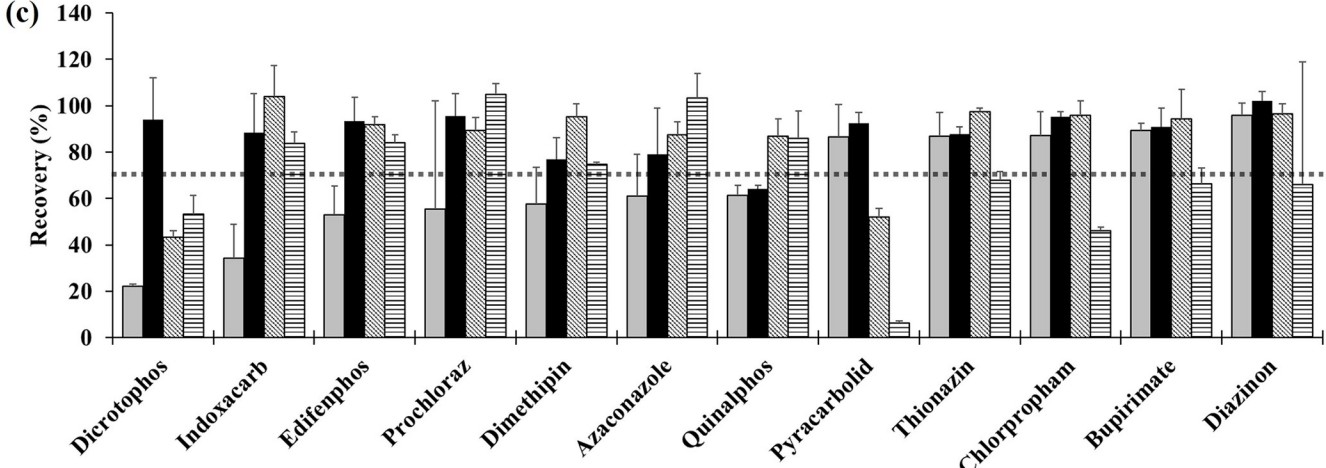

Pesticide in *Paeonia lactiflora*

□ ACN  ■ ACN/EA (7:3)  ▨ ACN/EA (3:7)  ▤ EA

**Fig 2.** Recoveries of representative pesticides showing a large recovery difference greater than 25% depending on extraction solvents which are acetonitrile (ACN), ACN/ethyl acetate (EA) (7:3, v/v), ACN/EA (3:7, v/v), and EA in *C. officinale* (a), *R. glutinosa* (b), and *P. lactiflora* (c). The error bars are the standard deviations of the recoveries (*n* = 3). The dotted lines mean the recovery of 70%.

and 2 pesticides, respectively, met the criteria. Similarly, 1, 9, 8, and 3 of the 10 compounds in *R. glutinosa* (Fig 2B) and 5, 11, 10, and 6 of the 12 in *P. lactiflora* (Fig 2C) showed recoveries of >70% when extracted with ACN, ACN/EA (7:3), ACN/EA (3:7), and EA, respectively. The ACN/EA mixtures showed superior extraction efficiency compared to ACN or EA alone. It seems that the intermediate polarity between ACN and EA is optimal for extracting pesticide multiresidues from these root/rhizome-based herbal medicines. In addition, ACN/EA (7:3) showed the lowest mean relative standard deviation (RSDs) of recovery rates for 296 pesticides in *C. officinale* and *R. glutinosa*, and the second lowest mean RSD in *P. lactiflora* (S2 Fig). In conclusion, ACN/EA (7:3) was selected as the optimum extraction solvent according to two conditions: lower EA ratio and better recovery rate.

In a further study, the recovery rates of the target pesticides according to the types and concentrations of acids in the optimized solvent were confirmed (S3 Fig). As a result, formic acid with 0.1% concentration showed the highest numbers of analytes satisfying the corresponding criteria. Lee et al. (2017) also reported that 0.1% formic acid in ACN exhibited better recovery rates for 360 GC-MS/MS amenable pesticides in brown rice than 1% formic acid or 0.1–1% acetic acid [26]. Recently, 0.1% formic acid in ACN has been selected as an alternative extraction solvent for QuEChERS procedures in crops [31], edible insects (mealworms) [18], and biological samples [24]. Our study confirmed that 0.1% formic acid was also effective in the ACN/EA (7:3, v/v) solvent for extracting multiple residues in root/rhizome-based herbal medicines.

## Optimization of sample cleanup

Samples extracted and partitioned with 10 mL of 0.1% formic acid in ACN/EA (7:3, v/v), 4 g of $MgSO_4$, and 1 g of NaCl were further purified using various methods, and their purification efficiencies were compared. In the recovery tests for 296 target pesticides, the percentages of analytes satisfying the excellent recovery range of 70–120% for each method were verified. As shown in Table 1, treatments with C18 (2) and alumina (7) sorbents in d-SPE and Oasis PRiME HLB plus light (8) showed higher percentages (81–89%) than others (69–77%) in all three samples. According to methods (3), (4), (5), and (6), the presence of PSA and GCB sorbents led to a decrease in the percentage of analytes. This indicates that some of the pesticides were adsorbed or removed by the PSA and GCB.

**Table 1. The percentages of 296 target pesticides satisfying recovery range 70–120% under various cleanup methods in *Cnidium officinale*, *Rehmannia glutinosa*, and *Paeonia lactiflora*.**

| Cleanup method | d-SPE sorbent or SPE type | % of analytes (%) | | |
|:---:|:---:|:---:|:---:|:---:|
| | | *C. officinale* | *R. glutinosa* | *P. lactiflora* |
| (1) | 25 mg PSA + 150 mg MgSO$_4$ | 77 | 75 | 74 |
| (2) | 25 mg C18 + 150 mg MgSO$_4$ | 86 | 87 | 86 |
| (3) | 25 mg PSA + 25 mg C18 + 150 mg MgSO$_4$ | 74 | 73 | 71 |
| (4) | 50 mg PSA + 50 mg C18 + 150 mg MgSO$_4$ | 69 | 69 | 68 |
| (5) | 25 mg PSA + 2.5 mg GCB + 150 mg MgSO$_4$ | 73 | 71 | 72 |
| (6) | 50 mg PSA + 7.5 mg GCB + 150 mg MgSO$_4$ | 71 | 70 | 71 |
| (7) | 25 mg alumina + 150 mg MgSO$_4$ | 88 | 89 | 87 |
| (8) | Oasis PRiME HLB cartridge plus light | 83 | 81 | 81 |

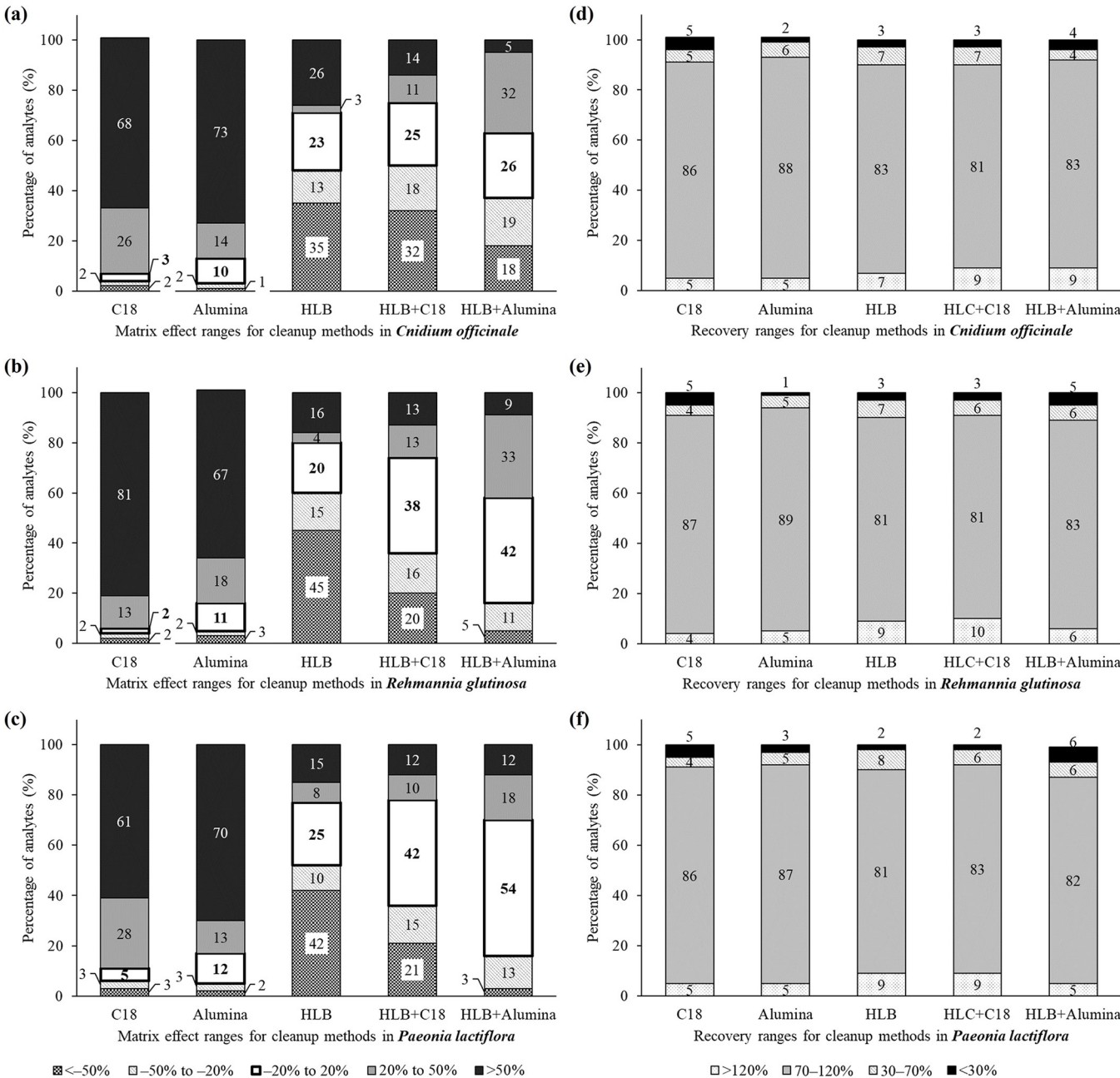

**Fig 3.** Distributions of matrix effects (% *ME*) for 296 target pesticides in *C. officinale*, *R. glutinosa*, and *P. lactiflora* (a)–(c), and distributions of recoveries for the same pesticides (d)–(f) under the various cleanup methods; C18 (25 mg C18 and 150 mg MgSO₄), Alumina (25 mg alumina and 150 mg MgSO₄), and HLB (Oasis PRiME HLB plus light).

When *% ME* was less than 0%, the signal was suppressed by the sample matrices. In contrast, a signal enhancement can be expected when the *% ME* is more than 0%. The matrix effect can be divided into three groups: soft effect (*% ME* between –20% and 20%), which is considered negligible; medium effect (*% ME* –50% to –20% or 20% to 50%); and strong effect (% ME <–50% or >50%) [32, 33]. In the single treatments of C18, alumina, and HLB, only 2–25% of the analytes exhibited *% ME* values within the soft effect range in the samples (Fig 3(A)–3(C)). In particular, most pesticides (61–81%) showed a strong signal enhancement of *% ME* >50%

in two d-SPE sorbents, whereas a large signal suppression of % *ME* <−50% was observed for most analytes (35–45%) in HLB. This means that the patterns of the matrix effects were considerably different between d-SPE and HLB. The determination of matrix effect patterns on GC-MS/MS is strongly related to the matrices remaining in the extracts [34]. Our study showed that each cleanup method removed different types of interferences, based on its specific purification mechanism.

To increase the removal efficiencies across a wide ranges of sample matrices, dual purifications were considered. This strategy enables the elimination of impurities that a single purification method cannot remove, by employing an additional purification procedure. Gong et al. (2020) demonstrated that the combining d-SPE and Oasis PRiME HLB resulted in superior matrix removal, compared to using HLB alone [35]. In this study, we implemented a dual purification strategy by conducting an Oasis PRiME HLB plus light cleanup, followed by d-SPE containing C18 or alumina sorbents.

As shown in Fig 3(A)–3(C), the percentages of analytes exhibiting the negligible matrix effect range (−20% to 20%) considerably increased in dual purifications (HLB+C18 and HLB+-Alumina; 25–54%) compared to single cleanup procedures (2–25%). Despite the stronger purification effect, the recovery results were similar to those of the single treatment with Oasis PRiME HLB plus light (Fig 3(D)–3(F)). Therefore, further co-elimination of the analytes by dual purification was not observed. Finally, we selected the combination of HLB and alumina as the established method because it reduced the strong matrix effect more than the HLB+C18 procedures.

## Method validation

In the established methodology, EA was added to the ACN solvent during extraction, which was found to improve the recovery of certain pesticides that were not adequately extracted with ACN alone. While this was beneficial for pesticide extraction efficiency, it increased the extraction of impurities or interferences. To counteract this, a dual purification process including Oasis PRiME HLB plus light and d-SPE containing alumina sorbents was employed, effectively minimizing pesticide loss and maximizing purification efficiency. The established analytical method for the 296 target pesticides underwent validation using three parameters, including LOQ, linearity of calibration, and recovery. The method was evaluated in accordance with SANTE/12682/2019 guidelines [36].

In the evaluation of sensitivity, 171 of 296 target pesticides (57.8% of the total) showed the lowest LOQ at 0.002 mg/kg (Tables 2 and 3). The sensitivity was similar to or better than the minimum LOQs reported in recent literatures (0.001–0.017 mg/kg), where more than 100 pesticides were simultaneously analyzed in root/rhizome-based herbal medicines [15, 16, 37, 38]. All of the analytes except ten met the LOQ requirement of 0.01 mg/kg or lower, indicating that the analytical method is suitable for national pesticide regulation systems where a default maximum residue level (MRL) of 0.01 mg/kg is normally applied when essential MRLs are lacking. The LOQs of the remaining ten pesticides did not exceed 0.05 mg/kg, indicating that the established method for three herbal medicines sufficiently determined targeted pesticide multiresidues with high sensitivity.

The linearity of the calibration curves, expressed as $r^2$, are shown in Table 3 as summarized data and in S2 Table as detailed values. The linear ranges of target analytes were between LOQ and 0.2 mg/kg. Most pesticides (73.6–82.8% of total) showed $r^2$ ≥0.990, indicating that their matrix-matched calibrations explain the correlation between concentration and signal well. Among the remaining pesticides, 26–42 compounds (8.8–14.2%) showed linearity $r^2$ between 0.980 and 0.990, which is suitable for multiresidue screening purposes. The number of analytes

**Table 2. Limit of quantitation (LOQ) and recovery rates fortified at 0.01 and 0.05 mg/kg for 296 target pesticides in _Cnidium officinale_, _Rehmannia glutinosa_, and _Paeonia lactiflora_.**

| No. (analyte) | Name | Cnidium officinale | | | Rehmannia glutinosa | | | Paeonia lactiflora | | |
|---|---|---|---|---|---|---|---|---|---|---|
| | | LOQ | Recovery, % (RSD, %) | | LOQ | Recovery, % (RSD, %) | | LOQ | Recovery, % (RSD, %) | |
| | | | 0.01 mg/kg | 0.05 mg/kg | | 0.01 mg/kg | 0.05 mg/kg | | 0.01 mg/kg | 0.05 mg/kg |
| 1 | 2,6-Diisopropyl- naphthalene | 0.005 | 103.9 (7.8) | 97.3 (13.0) | 0.005 | 119.5 (1.9) | 97.8 (5.6) | 0.005 | 96.4 (0.8) | 85.5 (1.8) |
| 2 | Acetochlor | 0.005 | 62.1 (0.3) | 58.5 (7.2) | 0.005 | 51.5 (2.6) | 61.8 (3.1) | 0.005 | 95.7 (1.3) | 83.9 (1.6) |
| 3 | EMA | 0.005 | 101.9 (3.3) | 102.2 (2.3) | 0.005 | 63.8 (2.6) | 34.5 (10.6) | 0.005 | 61.8 (1.3) | 68.5 (2.3) |
| 4 | HEMA | 0.002 | 85.3 (13.3) | 102.5 (5.3) | 0.002 | 91.5 (7.7) | 80.1 (5.7) | 0.002 | 95.6 (5.7) | 87.8 (18.2) |
| 5 | Acrinathrin | 0.002 | 96.3 (10.2) | 90.2 (8.4) | 0.002 | 68.5 (0.5) | 71.6 (0.7) | 0.002 | 93.6 (3.0) | 96.4 (5.0) |
| 6 | Alachlor | 0.005 | 93.6 (3.3) | 121.1 (10.5) | 0.005 | 101.8 (4.3) | 94.3 (2.3) | 0.005 | 95.2 (2.2) | 87 (2.8) |
| 7 | Aldrin | 0.002 | 99.2 (5.0) | 109.4 (13.5) | 0.002 | 89.6 (2.4) | 72.8 (6.9) | 0.002 | 88.7 (1.5) | 65.9 (19.4) |
| 8 | Dieldrin | 0.002 | 97.6 (6.1) | 95.2 (12.8) | 0.002 | 93.9 (4.0) | 93.8 (1.6) | 0.002 | 94.5 (4.9) | 89.8 (8.3) |
| 9 | Allidochlor | 0.005 | 96.1 (4.5) | 88.5 (2.7) | 0.005 | 97.5 (7.0) | 87.3 (16.6) | 0.005 | 71.5 (5.4) | 131.5 (15.5) |
| 10 | Ametryn | 0.002 | 105.9 (6.1) | 106.1 (6.0) | 0.002 | 108.8 (2.6) | 117.1 (11.5) | 0.002 | 75 (8.5) | 78.3 (3.5) |
| 11 | Anilofos | 0.005 | 101.7 (6.1) | 111.7 (5.2) | 0.005 | 55.5 (3.7) | 21.5 (2.4) | 0.005 | 98.6 (3.1) | 91.7 (3.3) |
| 12 | Aramite | 0.002 | 45.7 (15.5) | 58.6 (17.0) | 0.002 | 98.3 (1.4) | 91.8 (1.2) | 0.002 | 90.6 (0.7) | 94.3 (3.3) |
| 13 | Aspon | 0.005 | 89 (9.5) | 97.6 (18.3) | 0.005 | 82 (28.9) | 96.8 (2.7) | 0.005 | 87.4 (6.5) | 60.9 (10.3) |
| 14 | Atrazine | 0.002 | 97.2 (20.6) | 106.9 (18.5) | 0.002 | 107.4 (3.7) | 96 (3.7) | 0.002 | 95.5 (2.8) | 93.3 (0.6) |
| 15 | Azaconazole | 0.005 | 102.8 (4.2) | 99.8 (8.3) | 0.005 | 99.7 (2.9) | 91.1 (2.7) | 0.005 | 96.6 (0.6) | 94.5 (1.7) |
| 16 | Benfluralin | 0.002 | 91.7 (6.0) | 81.9 (2.0) | 0.002 | 95.3 (2.9) | 95.2 (7.4) | 0.002 | 97.3 (3.2) | 91.2 (4.5) |
| 17 | Benfuresate | 0.005 | 100.3 (13.2) | 87.5 (10.5) | 0.005 | 104.9 (9.0) | 105 (15.5) | 0.005 | 98.4 (2.1) | 79.5 (7.8) |
| 18 | Benodanil | 0.005 | 98.9 (3.8) | 96.5 (3.0) | 0.005 | 102 (2.5) | 90.7 (0.5) | 0.005 | 97.3 (1.7) | 92.3 (0.8) |
| 19 | Benoxacor | 0.005 | 50.2 (2.2) | 54.6 (12.5) | 0.005 | 51.5 (2.2) | 38.5 (0.9) | 0.005 | 42.9 (2.8) | 55.5 (1.1) |
| 20 | Benzoylprop-ethyl | 0.002 | 95 (5.5) | 100.2 (6.3) | 0.002 | 98.8 (2.1) | 87.1 (1.6) | 0.002 | 90.9 (3.2) | 87 (1.5) |
| 21 | BHC-alpha | 0.002 | 93.6 (0.7) | 91.2 (4.6) | 0.002 | 97.6 (2.1) | 95 (3.0) | 0.002 | 93.3 (2.7) | 88.2 (1.3) |
| 22 | BHC-beta | 0.002 | 96.2 (5.8) | 100.2 (6.4) | 0.002 | 98.9 (1.9) | 87.1 (1.6) | 0.002 | 91.4 (2.4) | 86.7 (1.6) |
| 23 | BHC-delta | 0.002 | 95.5 (13.3) | 71.5 (12.4) | 0.002 | 106.4 (3.4) | 100.3 (3.6) | 0.002 | 90.8 (2.1) | 99.2 (3.7) |
| 24 | BHC-gamma | 0.002 | 32.1 (7.0) | 62.9 (6.2) | 0.002 | 35.9 (6.8) | 48.5 (2.3) | 0.002 | 95.3 (3.1) | 95.7 (2.1) |
| 25 | Bifenox | 0.005 | 97.8 (6.2) | 75.2 (10.1) | 0.005 | 103.3 (6.9) | 88.9 (6.3) | 0.005 | 88.6 (1.5) | 86.4 (5.9) |
| 26 | Bifenthrin | 0.002 | 96.5 (11.5) | 103.1 (14.1) | 0.002 | 94.7 (4.8) | 91.8 (1.2) | 0.002 | 91.9 (4.0) | 104.1 (4.3) |
| 27 | Binapacryl | 0.005 | 97.6 (5.4) | 107.2 (2.2) | 0.005 | 107.4 (5.7) | 94.9 (3.4) | 0.005 | 97 (3.1) | 88.3 (1.9) |
| 28 | Boscalid | 0.002 | 94.5 (4.0) | 120.8 (9.3) | 0.002 | 99.5 (0.5) | 90.1 (1.3) | 0.002 | 92.8 (1.6) | 86.3 (1.9) |
| 29 | Bromobutide | 0.002 | 97.9 (6.7) | 97.4 (2.7) | 0.002 | 104 (3.8) | 94.4 (2.5) | 0.002 | 92.6 (0.2) | 97.1 (1.0) |
| 30 | Bromophos-ethyl | 0.002 | 97.8 (4.5) | 109.2 (13.9) | 0.002 | 97.7 (1.9) | 87.5 (2.9) | 0.002 | 94.3 (13.4) | 89.3 (10.3) |
| 31 | Bromophos-methyl | 0.005 | 88.4 (1.6) | 100.3 (8.8) | 0.005 | 112.2 (17.7) | 108.3 (11.6) | 0.005 | 90.5 (4.1) | 70 (9.1) |
| 32 | Bromopropylate | 0.002 | 105.1 (3.4) | 106.5 (12.3) | 0.002 | 85.9 (5.3) | 104 (8.6) | 0.002 | 91.8 (1.4) | 107.1 (15.5) |
| 33 | Bupirimate | 0.005 | 103 (6.3) | 83 (6.5) | 0.005 | 112.9 (4.8) | 103.3 (4.2) | 0.005 | 93.7 (1.7) | 98.7 (0.6) |
| 34 | Buprofezin | 0.002 | 95.6 (6.8) | 119.6 (6.8) | 0.002 | 98.2 (9.0) | 83.1 (7.4) | 0.002 | 93.3 (0.6) | 91.2 (2.2) |
| 35 | Butachlor | 0.005 | 43.5 (11.3) | 52.8 (10.6) | 0.005 | 65.8 (1.6) | 61.5 (11.2) | 0.005 | 96.4 (3.3) | 78 (11.6) |
| 36 | Butafenacil | 0.005 | 94.2 (5.9) | 102.1 (12.3) | 0.005 | 99.1 (2.0) | 87.5 (1.0) | 0.005 | 106.6 (3.9) | 89.6 (12.3) |
| 37 | Butralin | 0.002 | 98.2 (3.9) | 113.3 (7.6) | 0.002 | 106.1 (6.6) | 93.8 (3.0) | 0.002 | 93.4 (0.9) | 91.3 (0.9) |
| 38 | Butylate | 0.005 | 85 (2.2) | 87.2 (12.9) | 0.005 | 88.8 (3.8) | 85 (3.4) | 0.005 | 92.4 (1.1) | 95 (1.9) |
| 39 | Cadusafos | 0.002 | 102.5 (4.2) | 102.6 (4.8) | 0.002 | 101 (0.8) | 94 (2.4) | 0.002 | 93.8 (3.6) | 101.2 (3.4) |
| 40 | Carbophenothion | 0.002 | 88.3 (8.8) | 90.6 (6.1) | 0.002 | 40.9 (119.7) | 70.4 (12.5) | 0.002 | 102.6 (16.4) | 100.5 (0.7) |
| 41 | Carboxin | 0.002 | 96.6 (11.5) | 98.7 (4.1) | 0.002 | 101.9 (5.4) | 94.2 (2.6) | 0.002 | 95.7 (2.6) | 86.2 (4.0) |
| 42 | Carfentrazone-ethyl | 0.002 | 97.7 (4.7) | 106.7 (3.1) | 0.002 | 105.5 (2.8) | 96.5 (2.3) | 0.002 | 96.5 (4.3) | 91.6 (1.6) |
| 43 | Chinomethionat | 0.002 | 68.2 (8.7) | 47.5 (17.0) | 0.002 | 61.8 (9.1) | 58.5 (19.0) | 0.002 | 57.9 (2.7) | 58.5 (1.9) |
| 44 | Chlorbenside | 0.005 | 96.7 (11.2) | 113.5 (9.2) | 0.005 | 108.5 (6.6) | 101.3 (11.8) | 0.005 | 91.3 (8.7) | 91.7 (3.7) |

(_Continued_)

**Table 2.** (Continued)

| No. (analyte) | Name | *Cnidium officinale* | | | *Rehmannia glutinosa* | | | *Paeonia lactiflora* | | |
|---|---|---|---|---|---|---|---|---|---|---|
| | | LOQ | Recovery, % (RSD, %) | | LOQ | Recovery, % (RSD, %) | | LOQ | Recovery, % (RSD, %) | |
| | | | 0.01 mg/kg | 0.05 mg/kg | | 0.01 mg/kg | 0.05 mg/kg | | 0.01 mg/kg | 0.05 mg/kg |
| 45 | Chlorbufam | 0.002 | 37.5 (1.5) | 51.6 (9.5) | 0.002 | 71.5 (0.8) | 62.5 (3.6) | 0.002 | 93 (4.0) | 91.8 (2.9) |
| 46 | Chlordane | 0.002 | 95.6 (3.5) | 98.9 (6.5) | 0.002 | 96.9 (1.7) | 90.3 (2.1) | 0.002 | 95.1 (2.5) | 92.5 (2.5) |
| 47 | Chlorethoxyfos | 0.002 | 92.9 (4.4) | 105.8 (6.5) | 0.002 | 100.4 (1.5) | 95.5 (2.2) | 0.002 | 96.1 (1.2) | 100.9 (5.0) |
| 48 | Chlorfenapyr | 0.002 | 107.6 (9.7) | 111.2 (13.5) | 0.002 | 101.7 (4.9) | 94.1 (2.6) | 0.002 | 92.5 (2.3) | 89.7 (1.1) |
| 49 | Chlorfenson | 0.002 | 84.8 (5.0) | 86.1 (13.3) | 0.002 | 76.3 (6.2) | 79.2 (5.4) | 0.002 | 94.7 (6.5) | 85.9 (2.7) |
| 50 | Chlorflurenol-methyl | 0.002 | 103.6 (9.6) | 108.5 (11.6) | 0.002 | 104.2 (3.2) | 93.4 (0.6) | 0.002 | 94.7 (1.1) | 90.9 (2.1) |
| 51 | Chlornitrofen | 0.002 | 99.7 (16.5) | 90.7 (16.6) | 0.002 | 101 (2.2) | 95.3 (4.8) | 0.002 | 71.5 (9.7) | 77.2 (7.6) |
| 52 | Chlorobenzilate | 0.005 | 101.6 (4.2) | 111 (5.9) | 0.005 | 100.9 (1.4) | 89.3 (1.8) | 0.005 | 95.9 (6.5) | 87.3 (3.6) |
| 53 | Chloroneb | 0.005 | 97.4 (8.4) | 109.3 (13.7) | 0.005 | 121.4 (8.5) | 90.3 (5.9) | 0.005 | 97.6 (7.7) | 90.1 (10.8) |
| 54 | Chloropropylate | 0.002 | 91.1 (6.5) | 116.4 (4.9) | 0.002 | 101.7 (1.0) | 87.6 (3.4) | 0.002 | 95.3 (1.8) | 83.2 (4.5) |
| 55 | Chlorothalonil | 0.005 | 55.8 (2.7) | 62.4 (5.6) | 0.005 | 52.5 (1.1) | 58.5 (3.0) | 0.005 | 87.8 (4.2) | 95.1 (1.9) |
| 56 | Chlorpropham | 0.002 | 93.8 (2.4) | 107.2 (2.9) | 0.002 | 98.3 (7.6) | 85 (3.3) | 0.002 | 94 (3.6) | 100 (23.5) |
| 57 | Chlorpyrifos | 0.002 | 98.4 (1.5) | 112.3 (1.2) | 0.002 | 106.3 (4.2) | 87.6 (1.3) | 0.002 | 91.4 (3.0) | 92.5 (2.4) |
| 58 | Chlorpyrifos-methyl | 0.005 | 104.7 (5.9) | 100.8 (3.7) | 0.005 | 99.3 (0.3) | 91.8 (0.6) | 0.005 | 92.9 (2.2) | 89 (1.6) |
| 59 | Chlorthal-dimethyl | 0.002 | 95.4 (6.0) | 81.5 (7.4) | 0.002 | 101.4 (2.6) | 91.6 (2.3) | 0.002 | 97.4 (4.1) | 91.2 (1.7) |
| 60 | Chlorthion | 0.002 | 92.1 (21.8) | 86.5 (5.6) | 0.002 | 82.5 (7.5) | 77.7 (1.1) | 0.002 | 97.4 (9.9) | 104.9 (13.9) |
| 61 | Chlorthiophos | 0.002 | 105.5 (5.3) | 86.7 (6.9) | 0.002 | 94.6 (0.4) | 89.1 (3.9) | 0.002 | 91.9 (6.6) | 92.5 (4.2) |
| 62 | Chlozolinate | 0.002 | 102 (3.4) | 90.9 (9.8) | 0.002 | 95 (6.9) | 97.5 (9.8) | 0.002 | 98.7 (0.3) | 95.8 (0.6) |
| 63 | Cinidon-ethyl | 0.002 | 94.5 (11.8) | 82.4 (8.3) | 0.002 | 51.8 (3.4) | 38.5 (3.0) | 0.002 | 92 (2.8) | 88.2 (2.0) |
| 64 | Cinmethylin | 0.005 | 115.8 (25.6) | 107.6 (7.2) | 0.005 | 106 (2.4) | 102.8 (3.5) | 0.005 | 88.2 (5.7) | 76.6 (14.3) |
| 65 | Clomazone | 0.002 | 43.5 (7.9) | 38.5 (10.2) | 0.002 | 99.2 (5.5) | 100.5 (5.0) | 0.002 | 92.2 (1.6) | 100.8 (6.7) |
| 66 | Coumaphos | 0.002 | 103 (5.0) | 90.8 (5.8) | 0.002 | 100.4 (2.1) | 91.1 (9.3) | 0.002 | 93.3 (1.5) | 43.9 (2.8) |
| 67 | Cyanophos | 0.002 | 102.3 (7.7) | 95.9 (2.5) | 0.002 | 110.9 (8.3) | 101.1 (2.0) | 0.002 | 98.4 (2.7) | 110.5 (2.0) |
| 68 | Cyflufenamid | 0.002 | 96.5 (8.3) | 97 (11.7) | 0.002 | 91.3 (5.8) | 119.4 (3.9) | 0.002 | 104.6 (2.2) | 104.2 (2.8) |
| 69 | Cyfluthrin | 0.005 | 93.2 (6.2) | 81 (4.3) | 0.005 | 65.8 (1.3) | 61.9 (7.3) | 0.005 | 99.7 (9.0) | 97.2 (1.9) |
| 70 | Cyhalofop-butyl | 0.002 | 53.5 (4.2) | 56.8 (10.5) | 0.002 | 35.5 (1.2) | 28.5 (0.9) | 0.002 | 61.5 (11.8) | 68.6 (2.4) |
| 71 | Cyhalothrin | 0.002 | 94.8 (17.3) | 93.4 (11.5) | 0.002 | 94.6 (3.5) | 72.2 (19.0) | 0.002 | 68.5 (3.9) | 71.5 (7.0) |
| 72 | Cypermethrin | 0.005 | 62.8 (6.3) | 109.9 (3.8) | 0.005 | 107.6 (3.7) | 93 (1.0) | 0.005 | 94.4 (2.7) | 93.6 (1.6) |
| 73 | Cyprazine | 0.002 | 97 (3.3) | 115.7 (14.6) | 0.002 | 77.5 (2.0) | 66 (5.6) | 0.002 | 95.4 (0.3) | 85.9 (1.1) |
| 74 | Cyprodinil | 0.02 | nd [a] | 101.2 (15.3) | 0.02 | nd | 93.7 (2.9) | 0.02 | nd | 63.8 (3.0) |
| 75 | DDD (p,p) | 0.002 | 88 (2.2) | 104.7 (4.2) | 0.002 | 104 (4.2) | 92.6 (1.1) | 0.002 | 94.2 (1.1) | 103.6 (8.7) |
| 76 | DDE (p,p) | 0.005 | 73.5 (9.8) | 77.5 (5.7) | 0.005 | 85.5 (4.6) | 88.9 (16.6) | 0.005 | 64.4 (7.5) | 13.3 (7.6) |
| 77 | DDT (o,p) | 0.002 | 70.1 (8.1) | 110.5 (4.5) | 0.002 | 56.8 (18.9) | 71.5 (6.4) | 0.002 | 91.2 (9.8) | 94.3 (5.0) |
| 78 | DDT (p,p) | 0.005 | 56.8 (1.0) | 70.5 (5.6) | 0.005 | 90.2 (1.9) | 91.1 (1.2) | 0.005 | 92.6 (1.8) | 91.3 (3.0) |
| 79 | Deltamethrin | 0.005 | 53.5 (3.8) | 56.7 (6.4) | 0.005 | 68.6 (4.4) | 71.5 (2.4) | 0.005 | 68.5 (4.5) | 55.1 (2.5) |
| 80 | Tralomethrin | 0.002 | 38.5 (4.4) | 58.9 (10.0) | 0.002 | 87.3 (21.5) | 91.2 (2.7) | 0.002 | 32.5 (4.3) | 48.2 (7.9) |
| 81 | Desmetryn | 0.005 | 92.6 (6.4) | 98.2 (4.4) | 0.005 | 110.3 (2.8) | 79.3 (5.0) | 0.005 | 98.5 (0.9) | 88.1 (2.6) |
| 82 | Dialifos | 0.002 | 103.7 (5.6) | 113.3 (8.1) | 0.002 | 99.8 (2.0) | 93.1 (1.5) | 0.002 | 95.4 (1.9) | 89.6 (3.3) |
| 83 | Di-allate | 0.005 | 96.7 (10.0) | 90.7 (6.1) | 0.005 | 107.9 (0.1) | 103.6 (7.6) | 0.005 | 102.8 (5.7) | 110.7 (3.1) |
| 84 | Diazinon | 0.002 | 103.5 (9.2) | 118.5 (21.5) | 0.002 | 71.5 (6.0) | 81.5 (3.2) | 0.002 | 103.5 (2.7) | 90.6 (3.9) |
| 85 | Dichlobenil | 0.005 | 92.4 (3.6) | 89.2 (2.4) | 0.005 | 98.6 (2.2) | 87.5 (2.3) | 0.005 | 61.5 (4.8) | 55.8 (0.6) |
| 86 | Dichlofenthion | 0.002 | 99.9 (8.9) | 103 (4.0) | 0.002 | 105.4 (1.8) | 93.4 (1.8) | 0.002 | 95.5 (2.1) | 98.7 (0.7) |
| 87 | Dichlofluanid | 0.002 | 56.5 (5.0) | 42.5 (3.2) | 0.002 | 61.5 (4.4) | 51.8 (2.0) | 0.002 | 98.1 (0.8) | 95.2 (0.9) |
| 88 | Dichlormid | 0.002 | 95.4 (5.9) | 100.9 (11.5) | 0.002 | 106.2 (2.6) | 101 (2.6) | 0.002 | 105.4 (10.3) | 113.9 (2.6) |
| 89 | Diclobutrazol | 0.002 | 106.7 (11.2) | 98.7 (10.8) | 0.002 | 89.1 (11.9) | 75.8 (3.9) | 0.002 | 43.2 (7.0) | 22.1 (15.3) |

(*Continued*)

Table 2. (Continued)

| No. (analyte) | Name | *Cnidium officinale* | | | *Rehmannia glutinosa* | | | *Paeonia lactiflora* | | |
|---|---|---|---|---|---|---|---|---|---|---|
| | | LOQ | Recovery, % (RSD, %) | | LOQ | Recovery, % (RSD, %) | | LOQ | Recovery, % (RSD, %) | |
| | | | 0.01 mg/kg | 0.05 mg/kg | | 0.01 mg/kg | 0.05 mg/kg | | 0.01 mg/kg | 0.05 mg/kg |
| 90 | Diclofop-methyl | 0.002 | 97.5 (2.6) | 102.1 (3.7) | 0.002 | 101.5 (3.2) | 91.3 (4.3) | 0.002 | 94.2 (0.6) | 84.2 (3.3) |
| 91 | Dicloran | 0.005 | 95.1 (5.7) | 62.7 (12.0) | 0.005 | 92.4 (5.7) | 97.4 (8.9) | 0.005 | 98.7 (1.5) | 92.7 (1.2) |
| 92 | Dicofol | 0.002 | 51.5 (15.5) | 24.5 (9.5) | 0.002 | 55.3 (1.2) | 28.1 (2.1) | 0.002 | 35.8 (7.6) | 4.8 (5.2) |
| 93 | Dicrotophos | 0.005 | 92.9 (4.3) | 74.5 (9.1) | 0.005 | 95.6 (5.1) | 111.8 (13.9) | 0.005 | 97.7 (5.3) | 100.7 (2.0) |
| 94 | Diethatyl-ethyl | 0.005 | 101.9 (6.1) | 100.8 (1.8) | 0.005 | 103.5 (5.8) | 95.2 (1.7) | 0.005 | 93.2 (4.7) | 98.8 (2.9) |
| 95 | Diethofencarb | 0.005 | 104.6 (14.6) | 96.7 (8.1) | 0.005 | 102.5 (3.5) | 91.2 (10.4) | 0.005 | 90.3 (6.0) | 87.7 (3.0) |
| 96 | Difenoconazole | 0.002 | 84.8 (1.4) | 86.8 (14.6) | 0.002 | 101.9 (1.7) | 91.1 (2.6) | 0.002 | 92.9 (2.8) | 88.2 (0.4) |
| 97 | Diflufenican | 0.002 | 96.9 (6.0) | 101.1 (9.6) | 0.002 | 94.9 (3.1) | 89.1 (0.5) | 0.002 | 88.9 (1.9) | 92.5 (1.1) |
| 98 | Dimepiperate | 0.01 | 99.2 (1.4) | 98.1 (3.9) | 0.01 | 97.7 (1.3) | 92.4 (2.9) | 0.01 | 91.9 (1.0) | 91.4 (2.2) |
| 99 | Dimethachlor | 0.002 | 79.5 (2.8) | 89.9 (8.6) | 0.002 | 88.2 (7.1) | 79.3 (1.6) | 0.002 | 95.3 (5.9) | 57.5 (1.3) |
| 100 | Dimethametryn | 0.002 | 96 (5.6) | 73.4 (7.7) | 0.002 | 100.8 (1.2) | 91.3 (2.1) | 0.002 | 88 (7.4) | 97.8 (4.8) |
| 101 | Dimethenamid | 0.002 | 95 (6.5) | 72.1 (9.8) | 0.002 | 83.5 (19.1) | 83.3 (12.5) | 0.002 | 84.2 (6.2) | 91.8 (1.2) |
| 102 | Dimethipin | 0.002 | 95 (1.9) | 101.1 (4.6) | 0.002 | 128.5 (10.5) | 84.3 (5.9) | 0.002 | 91.6 (1.0) | 87.9 (0.9) |
| 103 | Dimethomorph | 0.002 | 96.2 (2.0) | 88.3 (3.5) | 0.002 | 98.8 (2.9) | 86.5 (2.9) | 0.002 | 91.4 (1.0) | 86.1 (1.1) |
| 104 | Dimethylvinphos | 0.002 | 68.5 (6.1) | 71.5 (1.4) | 0.002 | 58.5 (2.2) | 34.5 (1.7) | 0.002 | 68.9 (5.9) | 71.5 (1.8) |
| 105 | Diniconazole | 0.002 | 111.5 (10.6) | 113.4 (13.7) | 0.002 | 101.6 (3.7) | 93.1 (5.1) | 0.002 | 95.8 (6.5) | 96.9 (1.7) |
| 106 | Dinitramine | 0.002 | 98.3 (0.9) | 70.4 (4.8) | 0.002 | 103.1 (15.5) | 93 (6.9) | 0.002 | 73 (13.8) | 85.9 (8.6) |
| 107 | Dioxathion | 0.005 | 101.3 (4.4) | 99.4 (9.9) | 0.005 | 89.2 (9.6) | 81.4 (10.8) | 0.005 | 92.5 (2.1) | 95.8 (4.5) |
| 108 | Diphenamid | 0.005 | 91.2 (10.0) | 93.9 (10.5) | 0.005 | 102.7 (4.2) | 92.3 (1.2) | 0.005 | 99.1 (3.7) | 96.8 (1.5) |
| 109 | Diphenylamine | 0.005 | 105.2 (4.0) | 106.1 (4.9) | 0.005 | 104.8 (1.3) | 94.4 (2.6) | 0.005 | 92.8 (4.3) | 93.1 (0.4) |
| 110 | Dithiopyr | 0.002 | 91.5 (4.8) | 95.5 (17.3) | 0.002 | 105.3 (5.5) | 83.1 (5.3) | 0.002 | 91.8 (3.8) | 81.8 (4.0) |
| 111 | Edifenphos | 0.02 | nd | 99.4 (7.1) | 0.02 | nd | 97.6 (1.0) | 0.02 | nd | 118.5 (29.5) |
| 112 | Endosulfan-alpha | 0.005 | 95.3 (1.2) | 108.9 (1.8) | 0.005 | 106.8 (3.2) | 93.2 (2.2) | 0.005 | 96.3 (2.7) | 96.3 (2.1) |
| 113 | Endosulfan-beta | 0.002 | 93.1 (15.5) | 93.6 (7.5) | 0.002 | 81.5 (79.9) | 102.9 (1.5) | 0.002 | 95.4 (4.0) | 101.9 (12.0) |
| 114 | Endosulfan-sulfate | 0.005 | 91.9 (14.4) | 108.9 (10.1) | 0.005 | 102.3 (3.8) | 71.5 (3.1) | 0.005 | 100.5 (8.2) | 89.7 (8.2) |
| 115 | Endrin | 0.002 | 95.5 (9.3) | 94.7 (1.0) | 0.002 | 105.9 (11.4) | 93.3 (6.2) | 0.002 | 76.8 (6.7) | 110.1 (4.2) |
| 116 | Endrin-ketone | 0.002 | 93.7 (3.3) | 104.6 (8.3) | 0.002 | 99.1 (21.5) | 88.2 (3.7) | 0.002 | 90.7 (0.8) | 88.9 (0.7) |
| 117 | EPN | 0.005 | 98.5 (2.4) | 91.6 (6.7) | 0.005 | 107.7 (2.3) | 94.7 (2.0) | 0.005 | 93.4 (3.9) | 89.1 (4.1) |
| 118 | Epoxiconazole | 0.005 | 101 (11.7) | 79.5 (17.0) | 0.005 | 97.1 (0.7) | 88.1 (3.4) | 0.005 | 96.5 (3.5) | 98 (8.6) |
| 119 | EPTC | 0.002 | 92.6 (11.6) | 115.6 (19.5) | 0.002 | 98.5 (3.7) | 93.7 (1.2) | 0.002 | 96.1 (1.1) | 86.9 (5.5) |
| 120 | Etaconazole | 0.005 | 94.5 (7.1) | 99 (9.2) | 0.005 | 90.6 (28.0) | 56.1 (15.8) | 0.005 | 96.9 (6.2) | 93.7 (10.8) |
| 121 | Ethalfluralin | 0.002 | 99.6 (5.6) | 119.9 (10.5) | 0.002 | 106.1 (6.0) | 96.1 (4.4) | 0.002 | 92.6 (2.5) | 90.9 (1.2) |
| 122 | Ethion | 0.002 | 99.3 (6.1) | 113.4 (12.0) | 0.002 | 103.5 (3.1) | 97.7 (2.0) | 0.002 | 93.7 (1.0) | 96.1 (2.2) |
| 123 | Ethofumesate | 0.005 | 101.3 (21.5) | 106.1 (4.1) | 0.005 | 90.3 (2.0) | 93.5 (2.5) | 0.005 | 93.1 (8.2) | 89.3 (4.8) |
| 124 | Ethoprophos | 0.002 | 100.4 (6.1) | 96.6 (11.7) | 0.002 | 102.4 (3.5) | 74.1 (8.5) | 0.002 | 95.2 (1.6) | 83.9 (9.0) |
| 125 | Ethychlozate | 0.002 | 102.1 (3.2) | 102.7 (11.3) | 0.002 | 90.3 (4.6) | 107.9 (16.7) | 0.002 | 84.4 (5.5) | 91 (10.9) |
| 126 | Etoxazole | 0.002 | 102.4 (5.3) | 88.1 (10.1) | 0.002 | 86.6 (62.9) | 88 (17.1) | 0.002 | 93.4 (6.2) | 88.5 (1.5) |
| 127 | Etridiazole | 0.002 | 83.2 (4.3) | 106.5 (9.9) | 0.002 | 95.3 (5.4) | 81.1 (1.9) | 0.002 | 96.3 (2.2) | 82.2 (4.4) |
| 128 | Fenamidone | 0.005 | 95.5 (4.1) | 85.1 (4.8) | 0.005 | 98 (11.2) | 97.7 (14.3) | 0.005 | 89.5 (4.5) | 101 (9.7) |
| 129 | Fenarimol | 0.005 | 91.8 (2.9) | 79.8 (7.6) | 0.005 | 98.8 (2.5) | 105.2 (4.3) | 0.005 | 91.8 (2.2) | 96.1 (0.7) |
| 130 | Fenbuconazole | 0.005 | 93.8 (4.8) | 76.2 (5.8) | 0.005 | 105.4 (3.8) | 61.7 (17.5) | 0.005 | 91.4 (2.7) | 92.9 (2.3) |
| 131 | Fenchlorphos | 0.005 | 94.8 (4.7) | 113.9 (4.9) | 0.005 | 101.3 (1.0) | 93.8 (1.5) | 0.005 | 97.5 (1.8) | 85.1 (2.4) |
| 132 | Fenclorim | 0.005 | 94.9 (5.5) | 104 (7.0) | 0.005 | 98.8 (2.5) | 91.1 (1.8) | 0.005 | 91.8 (1.6) | 88.3 (1.8) |
| 133 | Fenfuram | 0.005 | 96.1 (12.7) | 95.5 (2.9) | 0.005 | 97.2 (1.3) | 88.9 (1.9) | 0.005 | 96.7 (0.5) | 89.6 (0.8) |
| 134 | Fenitrothion | 0.002 | 96.1 (4.1) | 103.9 (15.1) | 0.002 | 102.4 (1.7) | 92.6 (1.9) | 0.002 | 94.9 (3.9) | 87.6 (3.1) |

(Continued)

**Table 2.** (*Continued*)

| No. (analyte) | Name | *Cnidium officinale* | | | *Rehmannia glutinosa* | | | *Paeonia lactiflora* | | |
|---|---|---|---|---|---|---|---|---|---|---|
| | | LOQ | Recovery, % (RSD, %) | | LOQ | Recovery, % (RSD, %) | | LOQ | Recovery, % (RSD, %) | |
| | | | 0.01 mg/kg | 0.05 mg/kg | | 0.01 mg/kg | 0.05 mg/kg | | 0.01 mg/kg | 0.05 mg/kg |
| 135 | Fenobucarb | 0.05 | nd | 98.5 (4.3) | 0.05 | nd | 339.7 (14.3) | 0.05 | nd | 117.8 (4.2) |
| 136 | Fenothiocarb | 0.002 | 92.6 (5.3) | 97 (13.4) | 0.002 | 95.6 (2.9) | 79.9 (5.8) | 0.002 | 96.2 (1.7) | 89.9 (5.3) |
| 137 | Fenoxanil | 0.005 | 92.5 (1.9) | 68.5 (4.6) | 0.005 | 55.5 (2.3) | 43.9 (13.0) | 0.005 | 91.9 (7.4) | 66.7 (2.9) |
| 138 | Fenpropathrin | 0.01 | 102.6 (6.5) | 100.3 (3.5) | 0.01 | 99.5 (3.0) | 99 (0.8) | 0.01 | 94.2 (0.9) | 103.9 (2.5) |
| 139 | Fenpropimorph | 0.005 | 94.3 (8.1) | 96 (6.8) | 0.005 | 93.9 (5.6) | 82.3 (5.4) | 0.005 | 87.6 (7.5) | 98.4 (2.7) |
| 140 | Fenpyrazamine | 0.002 | 92.9 (8.5) | 78 (4.1) | 0.002 | 98.8 (1.1) | 100.4 (1.5) | 0.002 | 83.8 (3.5) | 77.7 (2.4) |
| 141 | Fenson | 0.005 | 92.4 (5.3) | 79.9 (8.4) | 0.005 | 98.3 (1.7) | 90.8 (3.9) | 0.005 | 89.1 (3.2) | 97.4 (3.1) |
| 142 | Fenthion | 0.002 | 100.1 (6.0) | 106.2 (7.0) | 0.002 | 100 (1.8) | 91.7 (1.5) | 0.002 | 92.9 (2.2) | 93.7 (1.6) |
| 143 | Fenvalerate | 0.005 | 97.2 (4.8) | 73.9 (9.4) | 0.005 | 104.5 (4.6) | 89.3 (9.4) | 0.005 | 96.4 (6.0) | 100.3 (23.2) |
| 144 | Fipronil | 0.01 | 96.6 (10.3) | 100.2 (4.7) | 0.01 | 101.9 (1.9) | 96.5 (1.7) | 0.01 | 99 (2.6) | 93.8 (1.1) |
| 145 | Flamprop-isopropyl | 0.002 | 113.9 (1.3) | 106.5 (4.3) | 0.002 | 98.1 (4.1) | 95.2 (3.2) | 0.002 | 100.7 (5.6) | 116.4 (1.6) |
| 146 | Fluacrypyrim | 0.002 | 96.3 (3.7) | 105.6 (7.5) | 0.002 | 99.6 (3.2) | 100.9 (9.5) | 0.002 | 99.4 (4.9) | 93.1 (2.1) |
| 147 | Fluazifop-butyl | 0.005 | 100.1 (0.5) | 104.4 (7.5) | 0.005 | 105.2 (4.3) | 99.8 (3.7) | 0.005 | 94.2 (1.6) | 62.4 (0.2) |
| 148 | Fluchloralin | 0.005 | 97 (14.8) | 123.8 (10.6) | 0.005 | 107.7 (2.2) | 95.9 (2.9) | 0.005 | 96.9 (2.7) | 94.7 (1.2) |
| 149 | Flucythrinate | 0.002 | 103.4 (6.1) | 74.4 (4.6) | 0.002 | 109.6 (3.2) | 100.7 (2.2) | 0.002 | 100.2 (1.9) | 118.6 (4.2) |
| 150 | Fluensulfone | 0.005 | 93.1 (8.0) | 88.4 (10.3) | 0.005 | 93.4 (3.2) | 93.2 (14.9) | 0.005 | 97.5 (17.1) | 60.8 (15.0) |
| 151 | Flufenpyr-ethyl | 0.002 | 101.3 (6.4) | 102.1 (12.7) | 0.002 | 100.4 (1.1) | 90.5 (2.9) | 0.002 | 94.7 (4.1) | 95.2 (3.3) |
| 152 | Flumetralin | 0.002 | 99.9 (3.7) | 117.7 (9.5) | 0.002 | 73.7 (80.0) | 98.3 (2.3) | 0.002 | 96 (1.5) | 92.7 (0.9) |
| 153 | Flumioxazine | 0.002 | 99 (10.5) | 81.2 (12.9) | 0.002 | 100.3 (4.9) | 93.9 (2.8) | 0.002 | 59.3 (14.8) | 70.3 (11.2) |
| 154 | Fluopyram | 0.002 | 96.4 (6.9) | 72.5 (9.6) | 0.002 | 96.2 (3.6) | 93.4 (2.9) | 0.002 | 94.8 (6.8) | 95.8 (3.0) |
| 155 | Flurochloridone | 0.002 | 106.1 (9.1) | 102.1 (2.4) | 0.002 | 94.2 (0.6) | 107.2 (1.2) | 0.002 | 92.3 (1.9) | 84.7 (6.5) |
| 156 | Fluorodifen | 0.005 | 95.8 (3.3) | 101.6 (3.9) | 0.005 | 108 (1.8) | 81.8 (15.2) | 0.005 | 100 (2.6) | 94.5 (1.5) |
| 157 | Fluquinconazole | 0.002 | 95 (5.1) | 78.7 (7.8) | 0.002 | 101.5 (1.5) | 91.7 (1.6) | 0.002 | 90.2 (2.9) | 99 (2.2) |
| 158 | Flurtamone | 0.002 | 104.4 (3.7) | 101.7 (7.6) | 0.002 | 103 (2.0) | 88 (0.7) | 0.002 | 94.7 (2.2) | 103.6 (0.7) |
| 159 | Flusilazole | 0.01 | 99.5 (6.6) | 86.5 (3.7) | 0.01 | 115.8 (23.2) | 87 (1.4) | 0.01 | 91.5 (1.8) | 98.3 (2.9) |
| 160 | Flutianil | 0.005 | 102.9 (4.6) | 96.3 (7.3) | 0.005 | 51.8 (11.5) | 27.5 (3.4) | 0.005 | 95.8 (7.7) | 90 (10.4) |
| 161 | Fluvalinate | 0.005 | 102.9 (3.4) | 79.2 (5.8) | 0.005 | 98.3 (1.8) | 91.4 (6.9) | 0.005 | 96.1 (7.1) | 89.5 (1.0) |
| 162 | Fluxapyroxad | 0.01 | 107.3 (11.5) | 83.3 (5.8) | 0.01 | 116.2 (2.0) | 99.8 (1.4) | 0.01 | 102.6 (6.8) | 101.6 (6.1) |
| 163 | Fonofos | 0.005 | 102.4 (7.9) | 117.8 (10.6) | 0.005 | 99.5 (1.8) | 92.3 (2.3) | 0.005 | 92.6 (1.8) | 87.8 (2.8) |
| 164 | Formothion | 0.002 | 68.5 (3.7) | 65.4 (11.9) | 0.002 | 51.5 (3.5) | 60.6 (4.0) | 0.002 | 61.5 (4.0) | 68.5 (1.1) |
| 165 | Fthalide | 0.002 | 103.5 (10.0) | 86.9 (6.8) | 0.002 | 108.6 (6.0) | 103.3 (1.3) | 0.002 | 94.8 (1.7) | 99.5 (4.6) |
| 166 | Halfenprox | 0.005 | 104 (3.8) | 87 (10.2) | 0.005 | 69.6 (76.5) | 75.6 (59.2) | 0.005 | 95.6 (7.7) | 115.2 (5.8) |
| 167 | Heptachlor | 0.005 | 91 (2.1) | 112.7 (7.3) | 0.005 | 102.7 (2.5) | 90.1 (1.1) | 0.005 | 91.4 (2.9) | 94 (2.9) |
| 168 | Heptachlor epoxide | 0.002 | 97.3 (4.1) | 111.8 (6.5) | 0.002 | 100.8 (5.2) | 92.8 (1.9) | 0.002 | 93.5 (5.1) | 87 (3.6) |
| 169 | Heptenophos | 0.002 | 97.7 (4.8) | 105.1 (11.4) | 0.002 | 98.7 (2.2) | 88.6 (2.0) | 0.002 | 94.1 (1.1) | 82.2 (1.4) |
| 170 | Hexachlorbenzene | 0.005 | 94.5 (2.1) | 110 (9.4) | 0.005 | 31.5 (2.2) | 28.6 (5.2) | 0.005 | 58.5 (7.7) | 51.6 (3.3) |
| 171 | Hexythiazox | 0.002 | 100.4 (4.8) | 90.1 (7.5) | 0.002 | 97.7 (30.7) | 103.8 (8.7) | 0.002 | 95.8 (3.2) | 99.8 (4.1) |
| 172 | Indanofan | 0.005 | 87.5 (4.5) | 102.1 (19.3) | 0.005 | 95.2 (1.6) | 71.5 (3.0) | 0.005 | 104 (12.7) | 34.1 (5.9) |
| 173 | Indoxacarb | 0.01 | 92.7 (23.5) | 98.5 (2.4) | 0.01 | 108 (21.9) | 86.8 (10.7) | 0.01 | 68.5 (1.3) | 71.5 (3.6) |
| 174 | Ipconazole | 0.005 | 98.2 (3.7) | 101.5 (11.3) | 0.005 | 104.6 (2.8) | 90.9 (3.7) | 0.005 | 95.7 (0.7) | 89.5 (2.6) |
| 175 | Iprobenfos | 0.002 | 97.9 (3.9) | 87.4 (4.2) | 0.002 | 99.1 (7.0) | 72.6 (6.4) | 0.002 | 95.8 (1.1) | 92.4 (6.1) |
| 176 | Iprodione | 0.002 | 92.7 (3.7) | 99.5 (3.6) | 0.002 | 98.9 (2.5) | 93.8 (2.3) | 0.002 | 92.3 (1.7) | 92.6 (4.0) |
| 177 | Isazofos | 0.002 | 103.5 (1.8) | 92.5 (6.2) | 0.002 | 99.8 (2.6) | 85.5 (3.7) | 0.002 | 100.1 (2.3) | 92.4 (4.1) |
| 178 | Isofenphos | 0.005 | 90.5 (10.9) | 104.8 (10.7) | 0.005 | 99.8 (1.5) | 94.8 (3.1) | 0.005 | 93.1 (2.4) | 89 (3.0) |
| 179 | Isofenphos-methyl | 0.005 | 100.4 (6.1) | 102.5 (0.6) | 0.005 | 109.5 (4.8) | 111.5 (2.1) | 0.005 | 93.3 (1.7) | 98.4 (1.5) |

(*Continued*)

**Table 2.** (Continued)

| No. (analyte) | Name | Cnidium officinale | | | Rehmannia glutinosa | | | Paeonia lactiflora | | |
|---|---|---|---|---|---|---|---|---|---|---|
| | | LOQ | Recovery, % (RSD, %) | | LOQ | Recovery, % (RSD, %) | | LOQ | Recovery, % (RSD, %) | |
| | | | 0.01 mg/kg | 0.05 mg/kg | | 0.01 mg/kg | 0.05 mg/kg | | 0.01 mg/kg | 0.05 mg/kg |
| 180 | Isoprocarb | 0.002 | 98.1 (8.5) | 108 (7.3) | 0.002 | 85.3 (6.0) | 102.9 (2.7) | 0.002 | 105 (14.0) | 95.7 (1.0) |
| 181 | Isopropalin | 0.002 | 104.5 (8.9) | 96.2 (4.1) | 0.002 | 81.9 (3.8) | 74.4 (6.0) | 0.002 | 100.1 (1.0) | 85.7 (4.7) |
| 182 | Isoprothiolane | 0.005 | 103.2 (5.5) | 86.1 (4.2) | 0.005 | 111.5 (1.2) | 112.2 (9.9) | 0.005 | 95.9 (2.6) | 100.8 (4.0) |
| 183 | Isopyrazam | 0.002 | 95.3 (6.9) | 90.7 (8.3) | 0.002 | 129.5 (29.5) | 91.3 (1.3) | 0.002 | 91.1 (3.0) | 92.7 (1.1) |
| 184 | Isotianil | 0.002 | 92.3 (12.3) | 100.9 (6.8) | 0.002 | 98.4 (2.2) | 95.1 (1.8) | 0.002 | 94.9 (4.6) | 92.2 (1.8) |
| 185 | Isoxadifen-ethyl | 0.002 | 106.3 (11.5) | 94.9 (2.6) | 0.002 | 114.6 (13.3) | 98.2 (1.9) | 0.002 | 107 (16.0) | 95 (2.9) |
| 186 | Kresoxim-methyl | 0.005 | 95.3 (1.9) | 96.1 (3.5) | 0.005 | 109.9 (5.6) | 91 (1.4) | 0.005 | 97.4 (1.6) | 94 (1.6) |
| 187 | Leptophos | 0.005 | 98.9 (6.3) | 96.7 (4.1) | 0.005 | 99.4 (1.4) | 92.8 (2.5) | 0.005 | 95.3 (1.6) | 94.2 (0.8) |
| 188 | Mefenpyr-diethyl | 0.002 | 100.9 (4.5) | 98.7 (3.6) | 0.002 | 94.4 (5.5) | 95.9 (8.5) | 0.002 | 96.3 (5.5) | 93.2 (1.1) |
| 189 | Mepanipyrim | 0.002 | 104.1 (6.5) | 92.1 (5.7) | 0.002 | 100.9 (3.6) | 95.8 (4.0) | 0.002 | 96.2 (4.8) | 99.3 (1.7) |
| 190 | Mepronil | 0.01 | 102.2 (25.8) | 83.5 (12.1) | 0.01 | 93.6 (3.6) | 92 (4.7) | 0.01 | 90.2 (3.2) | 94.7 (5.5) |
| 191 | Metalaxyl | 0.002 | 89.3 (10.0) | 94.6 (4.0) | 0.002 | 106.6 (4.6) | 73.6 (11.6) | 0.002 | 78.2 (25.2) | 83 (1.8) |
| 192 | Methidathion | 0.002 | 93.5 (12.0) | 90 (15.6) | 0.002 | 101.5 (5.1) | 108.5 (3.7) | 0.002 | 85.2 (15.9) | 96.1 (6.1) |
| 193 | Methoprotryne | 0.002 | 95.6 (5.8) | 110.4 (3.1) | 0.002 | 91.5 (3.1) | 96.8 (7.1) | 0.002 | 92.2 (3.5) | 112.1 (2.0) |
| 194 | Methoxychlor | 0.002 | 61.8 (13.4) | 62.5 (10.5) | 0.002 | 61.8 (0.4) | 68.5 (0.3) | 0.002 | 68.7 (4.0) | 65.3 (6.2) |
| 195 | Methyl trithion | 0.002 | 97.4 (7.1) | 103.3 (7.7) | 0.002 | 101.7 (2.3) | 92.9 (0.8) | 0.002 | 138.5 (15.6) | 96.3 (2.6) |
| 196 | Metolachlor | 0.002 | 43.4 (6.0) | 34.7 (13.3) | 0.002 | 73.7 (7.7) | 84.6 (5.2) | 0.002 | 81.3 (3.0) | 80.2 (3.7) |
| 197 | Metribuzin | 0.005 | 102 (3.0) | 101 (10.8) | 0.005 | 102.7 (7.8) | 93.6 (6.9) | 0.005 | 88.5 (3.1) | 82.2 (3.0) |
| 198 | MGK_264 | 0.002 | 91.7 (9.3) | 67.4 (8.6) | 0.002 | 102.4 (4.0) | 102.9 (5.8) | 0.002 | 83.8 (11.2) | 90 (3.6) |
| 199 | Mirex | 0.005 | 100.4 (1.2) | 108.6 (12.7) | 0.005 | 102.3 (1.4) | 93.5 (4.0) | 0.005 | 95.9 (1.3) | 84.6 (2.1) |
| 200 | Molinate | 0.005 | 75.4 (19.5) | 81.5 (5.5) | 0.005 | 88.5 (7.7) | 97.3 (7.4) | 0.005 | 96 (6.5) | 72.4 (13.7) |
| 201 | Monolinuron | 0.002 | 76.5 (21.5) | 101.9 (10.6) | 0.002 | 108 (6.0) | 79.4 (12.5) | 0.002 | 89.6 (6.8) | 90.3 (2.6) |
| 202 | Myclobutanil | 0.005 | 94.1 (5.8) | 110 (2.9) | 0.005 | 95.7 (11.9) | 82.4 (3.6) | 0.005 | 93.7 (10.6) | 96.9 (9.5) |
| 203 | Nitrapyrin | 0.002 | 97.1 (7.8) | 100.8 (14.5) | 0.002 | 119.7 (2.4) | 93.8 (5.7) | 0.002 | 94.7 (10.5) | 93.8 (1.1) |
| 204 | Nitrothal-isopropyl | 0.002 | 106.4 (2.1) | 111.8 (3.7) | 0.002 | 105.9 (5.8) | 98.5 (4.0) | 0.002 | 93.8 (4.0) | 92.3 (1.8) |
| 205 | Nonachlor | 0.002 | 91.9 (4.3) | 111.8 (1.9) | 0.002 | 105.8 (1.7) | 94.8 (3.5) | 0.002 | 88.7 (4.6) | 93.5 (1.0) |
| 206 | Nuarimol | 0.005 | 99.5 (6.3) | 86.7 (7.6) | 0.005 | 104.7 (3.5) | 94.6 (3.5) | 0.005 | 90.1 (9.8) | 97.6 (2.2) |
| 207 | O-Phenylphenol | 0.002 | 98.5 (4.4) | 103.6 (13.5) | 0.002 | 97.8 (1.7) | 90 (4.9) | 0.002 | 89.9 (3.6) | 90 (2.5) |
| 208 | Oxadiazon | 0.002 | 93.7 (13.9) | 90 (8.7) | 0.002 | 99.5 (4.6) | 89.9 (1.5) | 0.002 | 97.2 (7.4) | 94.2 (5.4) |
| 209 | Oxadixyl | 0.002 | 82.5 (5.2) | 92.4 (11.8) | 0.002 | 101 (21.5) | 92.6 (7.2) | 0.002 | 91.4 (4.7) | 87.5 (4.7) |
| 210 | Oxyfluorfen | 0.005 | 101.5 (6.4) | 90.8 (19.7) | 0.005 | 111.1 (3.7) | 95.6 (1.0) | 0.005 | 94.8 (2.2) | 94 (9.3) |
| 211 | Paclobutrazol | 0.002 | 96.5 (6.5) | 104.9 (5.0) | 0.002 | 86 (18.8) | 92.5 (1.3) | 0.002 | 96.1 (1.6) | 80.3 (2.9) |
| 212 | Parathion | 0.002 | 99.5 (3.7) | 107.9 (4.8) | 0.002 | 105.9 (0.4) | 90.4 (1.1) | 0.002 | 92.9 (3.9) | 93.1 (1.5) |
| 213 | Parathion-ethyl | 0.005 | 98.3 (3.0) | 94.7 (1.4) | 0.005 | 109.4 (4.5) | 95.8 (3.6) | 0.005 | 94.8 (1.5) | 98.5 (1.8) |
| 214 | Parathion-methyl | 0.005 | 88.4 (9.3) | 104.3 (11.8) | 0.005 | 99.6 (2.1) | 82.4 (7.5) | 0.005 | 91.7 (0.4) | 97.4 (2.7) |
| 215 | Penconazole | 0.02 | nd | 104.5 (4.9) | 0.02 | nd | 105.7 (1.5) | 0.02 | nd | 89.9 (3.5) |
| 216 | Pendimethalin | 0.01 | 106.3 (8.1) | 85.5 (4.4) | 0.01 | 97.6 (5.4) | 91.7 (4.9) | 0.01 | 93.8 (1.1) | 98.2 (3.8) |
| 217 | Penflufen | 0.002 | 87.9 (1.4) | 92.7 (6.3) | 0.002 | 98 (1.6) | 92.3 (1.2) | 0.002 | 94.3 (3.0) | 88.7 (1.8) |
| 218 | Pentachlorobenzonitrile | 0.002 | 109 (10.9) | 98.4 (5.4) | 0.002 | 97.8 (0.7) | 94.8 (2.6) | 0.002 | 97.8 (3.2) | 93.8 (2.9) |
| 219 | Penthiopyrad | 0.005 | 100.9 (5.6) | 97.9 (2.3) | 0.005 | 95.3 (1.8) | 95.7 (3.7) | 0.005 | 93.2 (3.5) | 100.6 (1.9) |
| 220 | Pentoxazone | 0.01 | 96.4 (4.2) | 84.8 (8.8) | 0.01 | 66.5 (7.6) | 102.9 (7.9) | 0.01 | 97.2 (2.4) | 99.7 (9.5) |
| 221 | Permethrin | 0.02 | nd | 86.1 (9.6) | 0.02 | nd | 99.4 (9.1) | 0.02 | nd | 97.1 (4.6) |
| 222 | Perthane | 0.01 | 104.7 (9.6) | 115.5 (5.7) | 0.01 | 108.1 (5.9) | 95 (0.9) | 0.01 | 95.2 (2.0) | 107.1 (3.1) |
| 223 | Phenthoate | 0.005 | 60.1 (9.2) | 62.8 (17.3) | 0.005 | 68.5 (2.7) | 68.1 (2.7) | 0.005 | 62.5 (1.4) | 55.9 (17.4) |
| 224 | Phosalone | 0.002 | 97.4 (5.9) | 92.7 (3.8) | 0.002 | 113.4 (21.5) | 81.2 (1.1) | 0.002 | 90.3 (5.0) | 92.6 (2.9) |

(Continued)

**Table 2.** (Continued)

| No. (analyte) | Name | *Cnidium officinale* | | | *Rehmannia glutinosa* | | | *Paeonia lactiflora* | | |
|---|---|---|---|---|---|---|---|---|---|---|
| | | LOQ | Recovery, % (RSD, %) | | LOQ | Recovery, % (RSD, %) | | LOQ | Recovery, % (RSD, %) | |
| | | | 0.01 mg/kg | 0.05 mg/kg | | 0.01 mg/kg | 0.05 mg/kg | | 0.01 mg/kg | 0.05 mg/kg |
| 225 | Phosmet | 0.01 | 98.9 (5.2) | 93.8 (6.7) | 0.01 | 101.6 (2.8) | 102.9 (6.0) | 0.01 | 83.2 (8.8) | 99.4 (12.1) |
| 226 | Phosphamidon | 0.002 | 67.5 (3.8) | 61.5 (2.8) | 0.002 | 67.5 (7.6) | 58.1 (4.4) | 0.002 | 68.6 (0.8) | 72.6 (6.8) |
| 227 | Picoxystrobin | 0.002 | 100.5 (5.5) | 93.2 (9.6) | 0.002 | 96.7 (10.3) | 71.8 (6.5) | 0.002 | 94.3 (21.0) | 61.2 (9.5) |
| 228 | Piperonyl butoxide | 0.002 | 95.3 (1.6) | 99.5 (6.5) | 0.002 | 108.3 (1.9) | 94.4 (1.6) | 0.002 | 94.9 (1.1) | 92.4 (0.7) |
| 229 | Pirimicarb | 0.002 | 104.1 (5.9) | 105.6 (7.5) | 0.002 | 99.6 (3.2) | 100.9 (9.5) | 0.002 | 99.4 (4.9) | 93.1 (2.1) |
| 230 | Pirimiphos-ethyl | 0.002 | 99.1 (2.2) | 102.4 (5.8) | 0.002 | 106.8 (7.0) | 96.8 (5.3) | 0.002 | 98.8 (8.7) | 91 (4.9) |
| 231 | Pirimiphos-methyl | 0.005 | 101.5 (8.1) | 91 (19.4) | 0.005 | 97 (1.5) | 89.2 (1.1) | 0.005 | 91.8 (2.3) | 85 (2.2) |
| 232 | Pretilachlor | 0.002 | 105.2 (8.2) | 103.9 (6.7) | 0.002 | 104 (2.5) | 91.2 (2.1) | 0.002 | 95.8 (2.9) | 85.2 (6.0) |
| 233 | Prochloraz | 0.002 | 51.5 (6.1) | 61.5 (13.7) | 0.002 | 61.8 (1.7) | 58.6 (5.1) | 0.002 | 93.5 (1.3) | 89.5 (1.5) |
| 234 | 2,4,6-Trichlorophenol | 0.002 | 38.1 (5.8) | 42.5 (8.0) | 0.002 | 98.1 (0.5) | 91.5 (6.1) | 0.002 | 130.2 (1.4) | 128.5 (10.8) |
| 235 | Procymidone | 0.005 | 53.8 (5.3) | 59.5 (7.3) | 0.005 | 77.7 (40.0) | 55.5 (17.5) | 0.005 | 89.2 (6.2) | 55.4 (4.6) |
| 236 | Prodiamine | 0.05 | nd | 101.7 (6.9) | 0.05 | nd | 79.3 (16.2) | 0.05 | nd | 89.9 (2.4) |
| 237 | Profenofos | 0.002 | 97.4 (9.1) | 109 (11.6) | 0.002 | 102.6 (3.6) | 94.9 (1.7) | 0.002 | 98 (2.1) | 93.8 (1.8) |
| 238 | Profluralin | 0.002 | 52.3 (3.9) | 107.9 (11.2) | 0.002 | 102 (0.9) | 88.8 (1.0) | 0.002 | 93.9 (2.2) | 90.2 (4.0) |
| 239 | Prohydrojasmon | 0.005 | 71.8 (5.3) | 81.5 (1.1) | 0.005 | 119.5 (21.8) | 99.7 (4.9) | 0.005 | 96 (1.4) | 100.3 (1.1) |
| 240 | Prometon | 0.002 | 88.1 (1.0) | 21.4 (9.6) | 0.002 | 79.5 (5.6) | 88.4 (1.3) | 0.002 | 89.7 (5.0) | 99.8 (11.1) |
| 241 | Prometryn | 0.002 | 89 (7.1) | 92 (2.9) | 0.002 | 90.5 (2.0) | 87.9 (3.2) | 0.002 | 94.4 (6.0) | 96.7 (3.0) |
| 242 | Propachlor | 0.002 | 98.3 (3.6) | 98.2 (10.7) | 0.002 | 92.8 (7.6) | 93.4 (0.9) | 0.002 | 95.1 (1.9) | 91 (1.9) |
| 243 | Propanil | 0.005 | 93.8 (21.8) | 99.5 (11.5) | 0.005 | 102.8 (3.0) | 90.6 (5.6) | 0.005 | 93.4 (4.5) | 82.9 (2.4) |
| 244 | Propazine | 0.05 | nd | 49.5 (7.2) | 0.05 | nd | 68.5 (3.6) | 0.05 | nd | 53.6 (14.9) |
| 245 | Propetamphos | 0.005 | 102 (3.8) | 109.4 (2.3) | 0.005 | 61.9 (1.9) | 71.5 (12.4) | 0.005 | 91.6 (31.2) | 88.6 (14.8) |
| 246 | Propham | 0.002 | 105.4 (5.9) | 105.5 (7.5) | 0.002 | 96.1 (5.9) | 89.5 (4.0) | 0.002 | 91.3 (3.7) | 93.8 (18.8) |
| 247 | Propiconazole | 0.005 | 51.5 (9.9) | 25.5 (3.1) | 0.005 | 129.5 (8.0) | 98.3 (3.6) | 0.005 | 104.3 (67.2) | 97.4 (10.5) |
| 248 | Propisochlor | 0.002 | 99.3 (7.0) | 98.1 (9.7) | 0.002 | 93.7 (8.8) | 83.5 (5.7) | 0.002 | 88.5 (8.8) | 91.9 (1.2) |
| 249 | Propyzamide | 0.005 | 98.7 (11.8) | 97.5 (1.2) | 0.005 | 106 (3.3) | 87.3 (13.1) | 0.005 | 96.7 (4.5) | 108.2 (13.2) |
| 250 | Prothiofos | 0.002 | 91 (14.5) | 104.2 (4.1) | 0.002 | 98.3 (2.2) | 93.8 (3.8) | 0.002 | 93.3 (2.7) | 92.3 (2.2) |
| 251 | Pyracarbolid | 0.002 | 95.7 (5.3) | 96.5 (13.7) | 0.002 | 106 (5.2) | 97.5 (1.4) | 0.002 | 93 (2.3) | 86.6 (4.3) |
| 252 | Pyraclofos | 0.01 | 95.8 (6.0) | 93.7 (9.6) | 0.01 | 99.5 (0.6) | 98.1 (1.9) | 0.01 | 91.5 (7.5) | 86.5 (15.4) |
| 253 | Pyraflufen-ethyl | 0.005 | 76.3 (9.5) | 85.5 (3.1) | 0.005 | 90.2 (14.3) | 80.2 (7.0) | 0.005 | 97.8 (27.1) | 111.1 (29.5) |
| 254 | Pyrazophos | 0.002 | 102 (4.9) | 107.9 (8.6) | 0.002 | 105.7 (1.0) | 95.5 (2.8) | 0.002 | 97.1 (1.5) | 96.7 (1.3) |
| 255 | Pyridalyl | 0.005 | 102 (12.5) | 89.7 (2.3) | 0.005 | 98.6 (2.6) | 89.4 (3.9) | 0.005 | 93.2 (2.5) | 102.7 (3.7) |
| 256 | Pyrifenox | 0.002 | 100 (4.7) | 86.4 (6.3) | 0.002 | 89.2 (6.1) | 109.8 (8.4) | 0.002 | 88.2 (6.5) | 87.7 (4.7) |
| 257 | Pyriftalid | 0.005 | 100.2 (5.2) | 96.2 (12.6) | 0.005 | 90.5 (3.5) | 85.8 (1.6) | 0.005 | 96.7 (0.6) | 89.8 (2.4) |
| 258 | Pyrimethanil | 0.002 | 98.1 (6.4) | 83.6 (4.8) | 0.002 | 125.9 (2.6) | 87.6 (1.7) | 0.002 | 91.2 (5.7) | 93.3 (4.1) |
| 259 | Pyriminobac-methyl | 0.002 | 86.3 (14.5) | 89.8 (10.1) | 0.002 | 99 (2.0) | 87.5 (4.2) | 0.002 | 89.7 (1.4) | 93.3 (3.8) |
| 260 | Quinalphos | 0.002 | 103 (9.7) | 101.6 (14.0) | 0.002 | 109.7 (6.4) | 92.3 (1.6) | 0.002 | 128.5 (2.5) | 92.8 (1.7) |
| 261 | Quinoxyfen | 0.002 | 96.1 (9.6) | 90.6 (4.8) | 0.002 | 123.9 (11.5) | 91.1 (2.9) | 0.002 | 96.1 (3.4) | 91 (1.0) |
| 262 | Quintozene | 0.002 | 82.4 (8.6) | 88.8 (2.7) | 0.002 | 96.6 (4.2) | 89.4 (11.3) | 0.002 | 86.7 (8.7) | 61.3 (13.8) |
| 263 | Quizalofop-ethyl | 0.002 | 92.1 (3.4) | 99.8 (6.2) | 0.002 | 99.5 (1.5) | 93.6 (1.0) | 0.002 | 90.2 (9.9) | 83.8 (1.2) |
| 264 | Silafluofen | 0.005 | 86.8 (9.4) | 99.5 (2.7) | 0.005 | 101 (3.4) | 92.6 (3.7) | 0.005 | 92.4 (1.5) | 88.4 (0.9) |
| 265 | Simeconazole | 0.005 | 98.9 (6.8) | 68.5 (9.1) | 0.005 | 103.4 (3.2) | 95 (0.9) | 0.005 | 94.3 (3.8) | 101 (1.5) |
| 266 | Simetryn | 0.002 | 100.9 (6.9) | 89 (9.0) | 0.002 | 108.5 (4.2) | 102.5 (1.0) | 0.002 | 102.3 (14.2) | 103.5 (3.4) |
| 267 | Spiromesifen | 0.002 | 102.4 (6.0) | 80.5 (1.7) | 0.002 | 110.6 (30.5) | 97.1 (3.9) | 0.002 | 90.7 (3.0) | 89.8 (1.3) |
| 268 | Spiroxamine | 0.002 | 51.5 (6.6) | 38.5 (11.8) | 0.002 | 59.6 (1.6) | 61.8 (3.3) | 0.002 | 61.8 (4.4) | 63.6 (6.8) |
| 269 | Sulfotep | 0.002 | 106.7 (0.9) | 105.8 (2.1) | 0.002 | 77.7 (58.3) | 73.1 (4.9) | 0.002 | 93 (3.4) | 87.8 (1.2) |

*(Continued)*

**Table 2.** (Continued)

| No. (analyte) | Name | *Cnidium officinale* | | | *Rehmannia glutinosa* | | | *Paeonia lactiflora* | | |
|---|---|---|---|---|---|---|---|---|---|---|
| | | LOQ | Recovery, % (RSD, %) | | LOQ | Recovery, % (RSD, %) | | LOQ | Recovery, % (RSD, %) | |
| | | | 0.01 mg/kg | 0.05 mg/kg | | 0.01 mg/kg | 0.05 mg/kg | | 0.01 mg/kg | 0.05 mg/kg |
| 270 | Tebuconazole | 0.002 | 99.7 (2.1) | 96.7 (6.1) | 0.002 | 89.6 (2.5) | 77.4 (2.3) | 0.002 | 101.7 (6.5) | 74 (4.2) |
| 271 | Tebufenpyrad | 0.002 | 99.9 (3.6) | 113 (6.6) | 0.002 | 100 (1.0) | 90.9 (1.9) | 0.002 | 92.2 (2.9) | 90.1 (0.5) |
| 272 | Tebupirimfos | 0.005 | 95.5 (7.4) | 85 (1.2) | 0.005 | 97 (2.2) | 85.8 (2.2) | 0.005 | 92.9 (0.3) | 94 (1.3) |
| 273 | Tecnazene | 0.005 | 102.7 (6.1) | 93.5 (1.1) | 0.005 | 102.2 (2.4) | 97.1 (1.6) | 0.005 | 98.2 (1.6) | 95.3 (1.8) |
| 274 | Tefluthrin | 0.002 | 95.3 (2.2) | 115.1 (5.1) | 0.002 | 102.6 (1.0) | 93.1 (3.8) | 0.002 | 95 (1.4) | 89.3 (5.4) |
| 275 | Terbacil | 0.002 | 97 (8.5) | 108.1 (6.8) | 0.002 | 102.4 (1.7) | 92.5 (2.9) | 0.002 | 127.6 (2.8) | 90 (0.4) |
| 276 | Terbumeton | 0.005 | 109.1 (21.5) | 86.5 (10.5) | 0.005 | 107.3 (3.3) | 94.3 (3.1) | 0.005 | 96.8 (2.3) | 92 (3.3) |
| 277 | Terbutryn | 0.002 | 96 (1.0) | 36.5 (12.9) | 0.002 | 101.9 (6.8) | 92.1 (1.9) | 0.002 | 91.4 (1.7) | 64.1 (11.8) |
| 278 | Tetrachlorvinphos | 0.002 | 87.2 (8.6) | 60.4 (14.7) | 0.002 | 93.3 (1.0) | 82.1 (0.8) | 0.002 | 112.7 (1.3) | 119.7 (3.6) |
| 279 | Tetraconazole | 0.005 | 102.3 (13.0) | 106.2 (6.2) | 0.005 | 99.8 (4.4) | 89 (3.2) | 0.005 | 97.6 (7.5) | 80.6 (3.4) |
| 280 | Tetradifon | 0.002 | 90.1 (2.3) | 102.4 (11.4) | 0.002 | 96.4 (5.3) | 86.3 (5.0) | 0.002 | 91.3 (5.0) | 83.8 (3.6) |
| 281 | Tetramethrin | 0.005 | 108.1 (8.0) | 101.5 (5.4) | 0.005 | 116.5 (21.5) | 89.6 (1.8) | 0.005 | 95.6 (5.4) | 89.7 (1.7) |
| 282 | Tetrasul | 0.002 | 100.1 (9.4) | 93.3 (4.6) | 0.002 | 61.8 (5.2) | 69.5 (0.5) | 0.002 | 75.7 (1.1) | 87.5 (3.3) |
| 283 | Thifluzamide | 0.005 | 96.6 (10.0) | 105.8 (9.9) | 0.005 | 41.6 (31.5) | 34.9 (8.9) | 0.005 | 97 (3.5) | 101.1 (3.7) |
| 284 | Thiometon | 0.05 | nd | 68.5 (6.7) | 0.05 | nd | 92.5 (1.1) | 0.05 | nd | 85.5 (2.3) |
| 285 | Thionazin | 0.005 | 94.6 (2.0) | 108.6 (8.7) | 0.005 | 107.8 (0.8) | 100 (2.1) | 0.005 | 101.4 (3.6) | 111.4 (2.8) |
| 286 | Tolclofos-methyl | 0.002 | 99.6 (5.8) | 52.9 (4.8) | 0.002 | 98.8 (6.6) | 85.3 (9.6) | 0.002 | 97.4 (6.0) | 98.4 (8.8) |
| 287 | Triadimefon | 0.002 | 100.2 (11.9) | 104.6 (3.9) | 0.002 | 98.9 (3.4) | 94.9 (6.5) | 0.002 | 97.3 (1.8) | 86.9 (5.4) |
| 288 | Triadimenol | 0.002 | 105.7 (9.2) | 99.2 (7.1) | 0.002 | 99.3 (3.4) | 94.6 (1.9) | 0.002 | 93.7 (1.2) | 97.4 (3.2) |
| 289 | Tri-allate | 0.05 | nd | 97.9 (2.8) | 0.05 | nd | 89.1 (7.6) | 0.05 | nd | 93.1 (6.0) |
| 290 | Triazophos | 0.002 | 100.9 (6.5) | 95.3 (5.6) | 0.002 | 100.8 (1.1) | 106.6 (2.2) | 0.002 | 86.9 (8.6) | 94.8 (7.8) |
| 291 | Tridiphane | 0.002 | 95.8 (26.5) | 111.7 (30.5) | 0.002 | 99.9 (1.1) | 90.4 (0.8) | 0.002 | 91.2 (6.2) | 88 (1.1) |
| 292 | Trifloxystrobin | 0.002 | 87.3 (11.8) | 109 (10.0) | 0.002 | 95.9 (2.5) | 96.5 (0.8) | 0.002 | 94.9 (8.7) | 105.5 (13.3) |
| 293 | Triflumizole | 0.05 | nd | 116.1 (17.7) | 0.05 | nd | 90.3 (16.9) | 0.05 | nd | 87 (6.3) |
| 294 | Trifluralin | 0.002 | 111.4 (11.1) | 95 (5.7) | 0.002 | 99 (3.4) | 102.1 (6.8) | 0.002 | 97.2 (5.7) | 97.7 (3.2) |
| 295 | Vinclozolin | 0.005 | 111.2 (5.6) | 94.7 (18.6) | 0.005 | 94.5 (5.1) | 91.3 (6.1) | 0.005 | 105.6 (2.3) | 99.1 (8.2) |
| 296 | Zoxamide | 0.005 | 61.5 (4.7) | 58.5 (14.2) | 0.005 | 51.5 (4.3) | 62.8 (4.1) | 0.005 | 97.3 (1.6) | 94.9 (2.4) |

[a]nd: not determined due to out of LOQ.

**Table 3. Summary of limit of quantitation (LOQ) and linearity of calibration ($r^2$) of the established analytical method for the 296 target pesticides in *Cnidium officinale*, *Rehmannia glutinosa*, and *Paeonia lactiflora*.**

| Validation parameter | Value | The number of pesticides satisfying the criteria | | |
|---|---|---|---|---|
| | | *Cnidium officinale* | *Rehmannia glutinosa* | *Paeonia lactiflora* |
| LOQ (mg/kg) | 0.002 | 171 (57.8) | 171 (57.8) | 171 (57.8) |
| | 0.005 | 103 (34.8) | 103 (34.8) | 103 (34.8) |
| | 0.01 | 12 (4.1) | 12 (4.1) | 12 (4.1) |
| | 0.02 | 4 (1.4) | 4 (1.4) | 4 (1.4) |
| | 0.05 | 6 (2.0) | 6 (2.0) | 6 (2.0) |
| | Sum | 296 (100) | 296 (100) | 296 (100) |
| Linearity ($r^2$) | $\geq 0.990$ | 218 (73.6) | 245 (82.8) | 238 (80.4) |
| | 0.980–0.990 | 42 (14.2) | 26 (8.8) | 33 (11.1) |
| | 0.900–0.990 | 36 (12.2) | 25 (8.4) | 25 (8.4) |
| | Sum | 296 (100) | 296 (100) | 296 (100) |

**Table 4. Summary of recovery results at 0.01 and 0.05 mg/kg of the established analytical method for the 296 target pesticides in *Cnidium officinale*, *Rehmannia glutinosa*, and *Paeonia lactiflora*.**

| Recovery (%) | RSD (%) | No. of pesticides (% of total) | | | | | |
|---|---|---|---|---|---|---|---|
| | | *Cnidium officinale* | | *Rehmannia glutinosa* | | *Paeonia lactiflora* | |
| | | 0.01 mg/kg | 0.05 mg/kg | 0.01 mg/kg | 0.05 mg/kg | 0.01 mg/kg | 0.05 mg/kg |
| <30 | ≤20 | 0 (0.0) | 3 (1.0) | 0 (0.0) | 5 (1.7) | 0 (0.0) | 3 (1.0) |
| | >20 | 0 (0.0) | 0 (0.0) | 0 (0.0) | 0 (0.0) | 0 (0.0) | 0 (0.0) |
| 30–70 | ≤20 | 29 (9.8) | 32 (10.8) | 30 (10.1) | 27 (9.1) | 20 (6.8) | 26 (8.8) |
| | >20 | 0 (0.0) | 0 (0.0) | 3 (1.0) | 0 (0.0) | 0 (0.0) | 0 (0.0) |
| 70–120 | ≤20 | 247 (83.4) | 256 (86.5) | 230 (77.7) | 262 (88.5) | 257 (86.8) | 261 (88.2) |
| | >20 | 10 (3.4) | 2 (0.7) | 17 (5.7) | 1 (0.3) | 5 (1.7) | 4 (1.4) |
| 120–140 | ≤20 | 0 (0.0) | 3 (1.0) | 5 (1.7) | 0 (0.0) | 4 (1.4) | 2 (0.7) |
| | >20 | 0 (0.0) | 0 (0.0) | 1 (0.3) | 0 (0.0) | 0 (0.0) | 0 (0.0) |
| >140 | ≤20 | 0 (0.0) | 0 (0.0) | 0 (0.0) | 1 (0.3) | 0 (0.0) | 0 (0.0) |
| | >20 | 0 (0.0) | 0 (0.0) | 0 (0.0) | 0 (0.0) | 0 (0.0) | 0 (0.0) |
| nd[a] | | 10 (3.4) | 0 (0.0) | 10 (3.4) | 0 (0.0) | 10 (3.4) | 0 (0.0) |
| Sum | | 296 (100) | 296 (100) | 296 (100) | 296 (100) | 296 (100) | 296 (100) |

[a]nd: not determined due to out of LOQ.

with $r^2 \geq 0.980$ was higher in *R. glutinosa* and *P. lactiflora* (271 each) than in *C. officinale* (260). This is because the matrices of *C. officinale* were found to be more complex than others based on the evaluation of the matrix effect (Fig 3(A)–3(C)) and the full scan (MS1) chromatograms of the control samples (S4 Fig). It appears that a larger amount of interferences in *C. officinale* reduced the linearity of some pesticides.

In recovery tests, 296 target compounds were subjected to fortification at 0.01 and 0.05 mg/kg. As shown in Tables 2 and 4, 230–257 pesticides (77.7–86.8% of the total) at 0.01 mg/kg and 256–262 (86.5–88.5%) showed an excellent recovery range of 70–120% and RSD ≤20%, respectively, based on the criteria of the SANTE guideline [36]. From the extraction efficiency study, it was verified that the recovery rates increased when using the mixed solvent of ACN and EA in a 7:3 volume ratio, as compared to the other solvents.

There are no available data for comparison of recovery values in the literature on pesticide multiresidues in *C. officinale*, *R. glutinosa*, and *P. lactiflora*, thus, other root/rhizome-based herbal medicines were verified and compared. For metalaxyl, paclobutrazol, and vinclozolin, Tang et al. (2006) compared two elution solvents (acetone and EA) using three matrix solid-phase dispersion (MSPD) sorbents (silica gel, florisil, and alumina) in *Isatis indigotica* Fort (dried root), and observed lower recovery ranges (55.5–76.7%) in alumina for both solvents than those in silica gel and florisil [39]. In this study, however, these pesticides did not lose their recoveries (73.6–111.2%) in our three herbal samples during the cleanup procedure with the same dispersive alumina (Table 2). It appears that the addition of ACN solvent during the extraction step reduced the adsorption of the analytes onto the alumina sorbent.

Compared to the ginseng root preparation with EA extraction followed by gel permeation chromatography (GPC) and SPE cartridge (GCB and PSA) [20], our analytical method showed higher recovery ranges for acrinathrin (≤25% vs. 68.5–96.4%) and dialifos (dialifor) (48–75% vs 89.6–113.3%) in all three samples (Table 2). Chlorothalonil and dichlofluanid showed similar recovery ranges in *C. officinale* and *R. glutinosa* (42.5–62.4%) compared to those in ginseng (33–75%), but these ranges were greater in *P. lactiflora* (87.8–98.1%). This indicates that some pesticides can obtain significantly different recovery ranges between samples, even when the same preparation method is applied.

Nine pesticides (benoxacor, chinomethionat, cyhalofop-butyl, dicofol, formothion, methoxychlor, phenthoate, propazine, and spiroxamine) showed lower recoveries (<70%) in all herbal medicines. Liu et al. (2016) reported that chinomethionat, cyhalofop-butyl, and phenthoate showed better recoveries (89.3–109.7%) in the roots and rhizomes of Chinese herbal medicines using ACN extraction followed by SPE as well as the AOAC 2007.01 Official QuEChERS Method [13, 15]. Yang et al. (2022) also reported higher recoveries (76.8–108.6%) for dicofol, phenthoate, and spiroxamine in *Panax notoginseng* when they used ACN extraction followed by d-SPE [38]. A common point in these preparation methods is ACN extraction, thus, it seems advantageous to use ACN for the extraction of these lower recovery pesticides from our herbal samples. In contrast, methoxychlor satisfied the acceptable recovery range in ginseng root when using EA as the sole extraction solvent [20].

## Application in herbal medicines

The established method has been applied to the herbal samples obtained from various commercial markets. Pesticide residues were verified in 47 samples originating from two sources: the Republic of Korea (32) and China (15). For samples from China, no pesticides were detected above the LOQ levels. In the samples from Korea, *P. lactiflora* did not contain any pesticides exceeding the LOQs, but some pesticides were detected in *C. officinale* and *R. glutinosa* (Table 5). In *C. officinale*, it was confirmed that at least one pesticide was detected in 12 of 14 samples, showing an 85.7% detection rate (S3 Table), and 10 pesticides were detected in these samples. Among them, difenoconazole, a triazole fungicide, was detected in 7 (50%) out of 14 samples, showing the highest detection frequency. For dimethomorph, the detection frequency (6; 42.9%) was relatively higher than that of the other pesticides, and the average residue (110.0 μg/kg) was the highest. In addition, this pesticide was solely detected in *R. glutinosa* (5 of 8 samples), and its average residue (170.1 μg/kg) was the highest among the samples. The individual quantitation results for pesticides in the samples are shown in S3 Table.

The residue patterns in the samples are similar to that reported in other studies. Yu et al. (2012) studied multiresidue determination in 11 kinds of root/rhizome samples in the Republic of Korea, and they found that the pesticide detection rate in *C. officinale* (75%) was higher than that in other kinds of samples (0–10%), including *P. lactiflora* (0%) [9].

**Table 5. Determination of pesticide multiresidues in *Cnidium officinale*, *Rehmannia glutinosa*, and *Paeonia lactiflora* which are origin from the Republic of Korea.**

| Sample | Pesticide'name | Classification | LOQ (μg/kg) | No. of detection (%) | Range (μg/kg) | Average (μg/kg) | Median (μg/kg) | Acceptable criteria (μg/kg) |
|---|---|---|---|---|---|---|---|---|
| *C. officinale* (14 samples) | Bifenthrin | Insecticide | 2 | 6 (42.9) | 10.5–31.5 | 23.0 | 24.3 | 600 |
| | Chlorpyrifos | Insecticide | 2 | 6 (42.9) | 15.2–41.2 | 26.2 | 25.0 | 600 |
| | Difenoconazole | Fungicide | 2 | 7 (50.0) | 21.5–52.2 | 33.3 | 32.1 | 600 |
| | Dimethomorph | Fungicide | 2 | 6 (42.9) | 50.2–150.5 | 110.0 | 114.7 | 12000 |
| | Diniconazole | Fungicide | 2 | 3 (21.4) | 30.9–33.3 | 32.4 | 32.9 | 138 |
| | Metribuzin | Herbicide | 5 | 3 (21.4) | 18.5–21.8 | 19.9 | 19.5 | 780 |
| | Pendimethalin | Herbicide | 10 | 5 (35.7) | 52.1–110.5 | 82.0 | 86.5 | 100 |
| | Quinalphos | Insecticide | 2 | 5 (35.7) | 10.5–24.5 | 16.8 | 15.8 | 50 |
| | Tebuconazole | Fungicide | 2 | 4 (28.6) | 21.5–52.5 | 33.9 | 30.7 | 1800 |
| | Tebufenpyrad | Insecticide | 2 | 3 (21.4) | 9.7–32.5 | 21.4 | 21.9 | 600 |
| *R. glutinosa* (8 samples) | Dimethomorph | Fungicide | 2 | 5 (62.5) | 58.2–321.5 | 170.1 | 100.5 | 4000 |
| *P. lactiflora* (10 samples) | Not detected in all samples. | | | | | | | |

In the Ministry of Food and Drug Safety of the Republic of Korea, acceptable limits of the pesticide residues in herbal medicines have been established based on the acceptable daily intake (ADI) and the daily intake of the corresponding herbal medicine [40]. Table 5 shows the acceptable criteria of the positively detected pesticides. Among the samples of *C. officinale*, only one showed a slightly higher level of pendimethalin at 110.5 µg/kg than the acceptable criteria of 100 µg/kg. However, all other detected pesticides exhibited lower concentrations than the acceptable criteria. The accumulation of monitoring studies can be used as a reference for conducting risk assessment in herbal medicines.

## Conclusions

A simultaneous analysis of 296 pesticides in three herbal medicines (*C. officinale*, *R. glutinosa*, and *P. lactiflora*) was developed with GC-MS/MS and modified QuEChERS method. Under the MRM detection mode and 15 psi pulsed-splitless injection of GC-MS/MS, extraction with acidified ACN/EA (7:3, v/v) and combination of Oasis PRiME HLB plus light and alumina d-SPE cleanup were found to be the optimal procedures for the multiresidue analysis in these herbal medicines. Using the established analytical method, we acquired reasonable validation data, including the LOQ (0.002–0.05 mg/kg), linearity of calibration, and recovery for most pesticides. The established method improved the extraction efficiency and reduced interferences, resulting in a reduction of the matrix effect for the target analytes. It was successfully applied to monitor multiresidues in samples obtained from commercial markets. The residue results can be used as reference data for the pesticide risk assessment in herbal medicines.

## Supporting information

**S1 Table. Retention times ($t_R$), multiple reaction monitoring (MRM) transitions, and collision energies (CEs) for 296 pesticides in GC-MS/MS.**
(PDF)

**S2 Table. The linearities of calibration curves expressed as $r^2$ for the target pesticides in herbal medicines.**
(PDF)

**S3 Table. Quantitation results of pesticide multiresidues in *C. officinale*, *R. glutinosa*, and *P. lactiflora* obtained from commercial markets.**
(PDF)

**S1 Fig. The average relative intensity (area) of target pesticides grouped by the four retention time ($t_R$) segments (8–14, 14–16.2, 16.2–18, and 18–25 min).** The average relative intensity in unpulsed injection was set to 100%.
(PDF)

**S2 Fig. The means of relative standard deviations (RSDs) for recoveries of 296 pesticides in *C. officinale*, *R. glutinosa*, and *P. lactiflora* under the extraction conditions of ACN, ACN/ EA (7:3, v/v), ACN/EA (3:7), and EA.** In cases where pesticides were not detected in certain methods and no RSD data was available, they were excluded from the statistics.
(PDF)

**S3 Fig. Distributions of recovery ranges of target pesticides when using 0.1, 0.4, and 1% formic acid or acetic acid in ACN/EA (7:3, v/v).**
(PDF)

**S4 Fig. Total ion chromatograms (TICs) through full scan analysis (m/z range 50–500).**
Control (pesticide-free) samples of (a) *C. officinale*, (b) *R. glutinosa*, and (c) *P. lactiflora* were
analyzed after preparation using the established method.
(PDF)

## Author Contributions

**Conceptualization:** Hoon Choi.

**Data curation:** Yongho Shin.

**Formal analysis:** Seung-Hyun Yang.

**Funding acquisition:** Hoon Choi.

**Investigation:** Seung-Hyun Yang.

**Methodology:** Seung-Hyun Yang, Yongho Shin.

**Project administration:** Hoon Choi.

**Resources:** Hoon Choi.

**Software:** Seung-Hyun Yang, Yongho Shin.

**Supervision:** Hoon Choi.

**Validation:** Seung-Hyun Yang.

**Visualization:** Yongho Shin.

**Writing – original draft:** Seung-Hyun Yang, Yongho Shin.

**Writing – review & editing:** Yongho Shin.

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
