## [Decision Letter · Decision Letter 0]

24 Jan 2023

PONE-D-22-33518Simultaneous analytical method for 296 pesticide multiresidues in root and rhizome based herbal medicines using GC-MS/MSPLOS ONE

Dear Dr. Choi,

Thank you for submitting your manuscript to PLOS ONE. After careful consideration, we feel that it has merit but does not fully meet PLOS ONE’s publication criteria as it currently stands. Therefore, we invite you to submit a revised version of the manuscript that addresses the points raised during the review process.

This article describes the development of an analytical method allowing the determination of 296 pesticides in medicinal plant samples by GC-MS/MS.

The objective of the study is clear and of major interest because the plants studied are used as medicines in Asian countries but also in Western countries.

This article has been evaluated by 2 independent reviewers (see reports below). In view of these different feedbacks, major revisions are requested for this article. In summary, this decision is based on 2 major aspects which are important criterias for publication in plos one:  

- methodologically: it is essential to better describe the "pesticide-free" reference matrix. How was it qualified as pesticide-free? What method was used to check this? How is it produced? These elements of precision are key technical points for the validation of the detection method.

- at the level of the discussion: this one will have to be deepened to put in perspective the measured levels of contaminants and/or residues with the literature, with the possible available reference values and/or with the possible regulatory values (maximum residual limits, acceptable daily dose). This reflection is essential to determine if the levels observed are significant and could have a health impact.

The remarks of the rapporteurs are detailed below and should all be taken into account in the revised article. If some of the reviewers' comments cannot be considered the authors should justify why this is not possible.

We look forward to receiving your revised manuscript.

Kind regards,

Christine Demeilliers

Academic Editor

PLOS ONE

Journal Requirements:

"This study was supported by a grant (21172MFDS149) from the Ministry of Food and Drug Safety in 2021."

"This study was supported by a grant (21172MFDS149) from the Ministry of Food and Drug Safety in 2021."

Reviewers' comments:

Reviewer's Responses to Questions

**Comments to the Author**

1. Is the manuscript technically sound, and do the data support the conclusions?

Reviewer #1: Yes

Reviewer #2: Partly

2. Has the statistical analysis been performed appropriately and rigorously? 

Reviewer #1: Yes

Reviewer #2: I Don't Know

3. Have the authors made all data underlying the findings in their manuscript fully available?

Reviewer #1: Yes

Reviewer #2: No

4. Is the manuscript presented in an intelligible fashion and written in standard English?

Reviewer #1: Yes

Reviewer #2: Yes

5. Review Comments to the Author

Reviewer #1: Please refer to the attached file for comment.

I used the tracking function.

Main Comment

The pesticide detection rate is very high.

Is PHI and MRL set for the detected pesticide?

The author should consider using this.

Reviewer #2: The authors describe an analytical method development dedicated to 296 pesticides in three herbal samples using GC-MS/MS. The experimental work is impressive and of good quality and focuses on: optimization of the extraction solvent, influence of complementary acidic conditions on the extraction process, further clean-up of the samples. The purpose of this study is of a great interest as it is suggested that the considered herbals are traditionally used as medicines and have become popular in Western countries.

Therefore, the introduction should describe in more details 1) the choice of the herbal medicines, 2) the choice of the pesticides studied, 3) the rules related to the use of such medicines at least in China and Korea. For example, the maximum residue limits are said to be slightly different in herbal medicines than in common crop but not given. Moreover even if no official rule is said to regulate the targeted compounds, an order of magnitude of the acceptable daily intake of pesticides residues according to the local pharmacopeia would be appreciated.

When developing the choice of the extraction solvents, more details on the chemical composition of herbal matrix is needed: why is it so different from common crops? Acetonitrile is considered as an extraction solvent used in the QuEChERS method whereas strictly speaking it does not allow two phases in an aqueous sample. The mixture of acetonitrile with an extraction solvent chose as ethyl acetate is expected to combine strength of both solvents: a reference or explanation is needed here.

The material and method part is well described but a doubt remains considering the specificity of the method: the “pesticide-free” herbal medicines are used as a blank matrix. How are these herbal medicines produced? How is the absence of all the 296 pesticides determined at the level of sensitivity described later in the paper. It suggests that the company from which these pesticide-free were purchased is able to quantify the targeted pesticides. The experimental conditions used for the GC-MS/MS analysis should be referenced.

The results and discussion part is clear and pertinent but:

- p10 - The choice of the ACN/EA ratio should be documented: why 3/7 and not 4/6, 5/5? Is it based from previous work?

- p11 - How does the study show that EA is more likely to co-extract interferences that ACN?

- Figures are blurred and red could not be seen apart from SI ones

- p11 - Tiometon is described but not visible from fig. 2

- figS2 shows a mean relative standard deviation over the 296 pesticides but no error bars are seen

- p15 - The method validation should be referenced or described in more details as only the performances in terms of LOQ is here given. If the MRL are available for some compounds, they should be given or referenced.

- p22 – is FigS4 meant after the whole treatment?

The method is finally applied to samples obtained from commercial markets with nice performances. It highlights that and the 296 targeted compounds, a maximum of 10 where detected beyond the LOQ in of the herbal medicine. More details would be appreciated here about the origin of the samples: not only the country but also the method of cultivation, of sampling, of storage...are these results of concern in terms of health risk?

6. PLOS authors have the option to publish the peer review history of their article (what does this mean?). If published, this will include your full peer review and any attached files.

Reviewer #1: No

Reviewer #2: No

---

## [Author Response · Author response to Decision Letter 0]

21 Feb 2023

February 21, 2023

Christine Demeilliers

Academic Editor

PLoS One

Dear Editor:

Thank you very much for valuable comments to our manuscript (PONE-D-22-33518) titled “Simultaneous analytical method for 296 pesticide multi-residues in root and rhizome based herbal medicines using GC-MS/MS”, coauthored by Seung-Hyun Yang and Yongho Shin. We sincerely and carefully revised manuscript according to editor and reviewers’ comments.

Revised parts were highlighted in red color as well as tracked changes. Those valuable comments greatly helped to improve the quality of this revised manuscript. We believe that all these revisions make our manuscript to be acceptable for the publication. Followings are our point-by-point responses to academic editor’s comments.

This study was supported by a grant (21172MFDS149) from the Ministry of Food and Drug Safety for the draft submission, and there was no additional external funding received for this study during revision period. The funders had no role in study design, data collection and analysis, decision to publish, or preparation of the manuscript.

It will be a great honor for us that our manuscript is accepted in the prestigious journal PLoS One.

Sincerely,

Hoon Choi

Department of Life & Environmental Sciences

College of Agriculture and Food Sciences

Wonkwang University, Iksan, 54538, Republic of Korea

Phone number: +82-63-850-6678

Fax number: +82-63-850-7308

Email address: hchoi0314@wku.ac.kr

 

<Author’s Response to Reviewer Comments for “PONE-D-22-33518”>

Academic Editor:

1. methodologically: it is essential to better describe the "pesticide-free" reference matrix. How was it qualified as pesticide-free? What method was used to check this? How is it produced? These elements of precision are key technical points for the validation of the detection method.

Answer: Thank you for your valuable comments. We were unable to confirm from the manufacturer how it was produced, but we confirmed the absence of pesticides in the samples using three established versions of the QuEChERS analysis method. We included this information in the Materials and Methods section (Lines 117-118 in Track Changes version). “Pesticides were confirmed to be absent in these samples using three different versions of the QuEChERS method [13, 21, 23].”

2. at the level of the discussion: this one will have to be deepened to put in perspective the measured levels of contaminants and/or residues with the literature, with the possible available reference values and/or with the possible regulatory values (maximum residual limits, acceptable daily dose). This reflection is essential to determine if the levels observed are significant and could have a health impact.

Answer: Thank you for your insightful feedback. For MRL, it is not established for the three herbs in the Republic of Korea nor China. Therefore, we used database of acceptable criteria of the Ministry of Food and Drug Safety of the Republic of Korea, which is based on the ADI and the daily intake of the corresponding herbal medicine. As a result, most of pesticides in the samples showed lower levels than those of acceptable criteria. The residue results can be used as reference data for the pesticide risk assessment in herbal medicines. We described this information in Lines 425-431 of the manuscript.

Reviewer #1:

1. Please refer to the attached file for comment. I used the tracking function.

Answer: Thank you for your valuable comments. Based on your suggestion, we correct the manuscript and revised parts were highlighted in red color as well as tracked changes.

2. Main Comment: The pesticide detection rate is very high. Is PHI and MRL set for the detected pesticide? The author should consider using this.

Answer: Thank you for your valuable comments. In the Ministry of Food and Drug Safety of the Republic of Korea, acceptable criteria of the pesticide residues in herbal medicines have been established based on the acceptable daily intake (ADI) and the daily intake of the corresponding herbal medicine. We showed the acceptable criteria in Table 5 and found that residue levels of most of pesticides were below acceptable criteria. We described this information in Lines 425-431 of the manuscript (Track Changes version).

Reviewer #2: 

1. The introduction should describe in more details 1) the choice of the herbal medicines, 2) the choice of the pesticides studied, 3) the rules related to the use of such medicines at least in China and Korea. For example, the maximum residue limits are said to be slightly different in herbal medicines than in common crop but not given. Moreover even if no official rule is said to regulate the targeted compounds, an order of magnitude of the acceptable daily intake of pesticides residues according to the local pharmacopeia would be appreciated.

Answer: Thank you for your valuable comments. We described the answer to question in the introduction. 1) the choice of the herbal medicines: Herbal medicines are commonly used in many countries, and the most popular ones in the Republic of Korea are Cnidium officinale, Rehmannia glutinosa, and Paeonia lactiflora. We have already fully explained the reasons for choosing three herbal medicines at Lines 45-52 (Track Changes version). 2) the choice of the pesticides studied: We selected the target pesticides because Most of the targeted pesticides are regulated as MRLs in food in both the Republic of Korea and China. (Lines 88-89) 3) the rules related to the use of such medicines at least in China and Korea: We also include the MRLs of China in Lines 58-60. ADI information of target pesticides is available on the URL:

https://www.mfds.go.kr/brd/m_1060/view.do?seq=14475&srchFr=&srchTo=&srchWord=&srchTp=&itm_seq_1=0&itm_seq_2=0&multi_itm_seq=0&company_cd=&company_nm=&page=1

We presented the acceptable criteria for pesticides in herbal medicines based on the ADI in Lines 425-431 of the manuscript.

2. When developing the choice of the extraction solvents, more details on the chemical composition of herbal matrix is needed: why is it so different from common crops?

Answer: Thank you for your insightful feedback. Roots and rhizomes of herbal medicines contain more abundant complex matrices that contain many biologically active ingredients, secondary metabolites, and phytochemicals than other foods, however, the co-extraction of these phytochemicals with pesticides during sample preparation can interfere with the accurate determination of pesticide residues. We have added this sentence to manuscript with a reference in Lines 65-71.

3. Acetonitrile is considered as an extraction solvent used in the QuEChERS method whereas strictly speaking it does not allow two phases in an aqueous sample.

Answer: Acetonitrile is a polar solvent that exhibits miscibility with water. However, the addition of sodium chloride (NaCl) increases the polarity of water, leading to phase separation from acetonitrile. QuEChERS contains NaCl as a component.

4. The mixture of acetonitrile with an extraction solvent chose as ethyl acetate is expected to combine strength of both solvents: a reference or explanation is needed here.

Answer: ACN and EA are well-miscible, and by adjusting their ratio, a desired polar solvent mixture can be obtained. As the reviewer’s valuable opinion, we included the sentence with a reference in Lines 81-82.

5. The material and method part is well described but a doubt remains considering the specificity of the method: the “pesticide-free” herbal medicines are used as a blank matrix. How are these herbal medicines produced? How is the absence of all the 296 pesticides determined at the level of sensitivity described later in the paper. It suggests that the company from which these pesticide-free were purchased is able to quantify the targeted pesticides.

Answer: Thank you for your valuable comments. We purchased pesticide-free herbal samples from a company, so we do not know the producing process of these samples. However, we confirmed the absence of pesticides in these samples using three existing representative QuEChERS methods. We have described this in Lines 117-118 of the manuscript.

6. The experimental conditions used for the GC-MS/MS analysis should be referenced.

Answer: We modified the GC-MS/MS conditions based on the method of Park et al. (2022), and this information has been included with the reference in Lines 134-135 of the manuscript.

7. p10 - The choice of the ACN/EA ratio should be documented: why 3/7 and not 4/6, 5/5? Is it based from previous work?

Answer: Thank you for your insightful feedback. We attempted to find a study on the QuEChERS extraction efficiency of ACN and EA mixture, but there were limited resources available. This study is believed to be the first attempt to compare the extraction solvent using the ACN/EA mixture. Therefore, we carried out the experiment with the four solvent conditions of ACN 100%, ACN/EA (7:3), ACN/EA (3:7), and EA 100% by adjusting the ratio of ACN and EA by 30-40% difference. While it is possible to conduct more detailed experiments with different ratios of ACN and EA, such as 10:0, 8:2, 6:4, 4:6, 2:8, and 0:10, this would greatly increase the number of experiments and it is also expected that there is not much difference between adjacent conditions.

8. p11 - How does the study show that EA is more likely to co-extract interferences that ACN?

Answer: We extracted control samples using ACN and EA and measured the weight of dry matters after drying the crude extracts. We found that in all three types of herbal samples, the weight of the dry matters from EA extract was greater than that from the ACN extract, confirming that co-extract was highly contained. We have summarized this result and presented it in Fig 1.

9. p11 - Figures are blurred and red could not be seen apart from SI ones.

Answer: Thank you for your valuable comments. In Fig 2, we increased the size of the text and used bold font and changed the red dotted line to black dotted line to make it more visible.

10. p11 - Tiometon is described but not visible from fig. 2

Answer: As the reviewer’s valuable mention, Thiometon was added in Fig 2 a. We have included "Fig 2 a, Fig 2 b, and Fig 2 c" in the sentences on Lines 249 and 252 to make it clear which parts of the figures the sentences are explaining.

11. figS2 shows a mean relative standard deviation over the 296 pesticides but no error bars are seen

Answer: Yes, Fig S2 shows a mean RSD for recoveries of the 296 pesticides. The RSD values of the individual pesticides are varied widely. Even some pesticides were not detected in certain methods, thus excluded from the statistics. Therefore, the error bars were not used in this figure, so they are included in the supporting information to verify the overall trends of repeatability for each preparation. In order to provide a clear explanation of the figure, we have added the sentence "In cases where pesticides were not detected in certain methods and no RSD data was available, they were excluded from the statistics." at the end of the caption.

12. p15 - The method validation should be referenced or described in more details as only the performances in terms of LOQ is here given. If the MRL are available for some compounds, they should be given or referenced.

Answer: As the reviewer’s valuable mention, we have included the following sentence along with the reference (SANTE/12682/2019) for the analytical method in Lines 326-329: “The established analytical method for the 296 target pesticides underwent validation using three parameters, including LOQ, linearity of calibration, and recovery. The method was evaluated in accordance with SANTE/12682/2019 guidelines [30].”

13. p22 – is FigS4 meant after the whole treatment?

Answer: Yes, FigS4 is typically meant to be viewed after the entire treatment. In order to provide a clear explanation of the figure, we have added the words "after preparation using the established method." at the end of the caption.

14. The method is finally applied to samples obtained from commercial markets with nice performances. It highlights that and the 296 targeted compounds, a maximum of 10 where detected beyond the LOQ in of the herbal medicine. More details would be appreciated here about the origin of the samples: not only the country but also the method of cultivation, of sampling, of storage...are these results of concern in terms of health risk?

Answer: Thank you for your valuable opinion. We were unable to obtain information about the cultivation, sampling, and storage of herbal samples as we purchased all the samples from the commercial markets. The aim of this study is to apply the novel analytical method to real samples, and the method will be used as a standard method in residue studies of crops to establish MRLs and PHIs of pesticide in herbal samples.

---

## [Decision Letter · Decision Letter 1]

2 May 2023

PONE-D-22-33518R1Simultaneous analytical method for 296 pesticide multi-residues in root and rhizome based herbal medicines with GC-MS/MSPLOS ONE

Dear Dr. Choi,

Thank you for submitting your manuscript to PLOS ONE. After careful consideration, we feel that it has merit but does not fully meet PLOS ONE’s publication criteria as it currently stands. Therefore, we invite you to submit a revised version of the manuscript that addresses the points raised during the review process.

We look forward to receiving your revised manuscript.

Kind regards,

Totan Adak

Academic Editor

PLOS ONE

Journal Requirements:

Additional Editor Comments:

I have assessed the manuscript entitled "Simultaneous analytical method for 296 pesticide multi-residues in root and rhizome based herbal medicines with GC-MS/MS". Please find my comments below in addition to the comments of the reviewers.

Following are major suggestions:

1. Thorough English editing is needed. Difficult to follow the write up. For example, see the line number 29-30 in abstract. I am unable to comprehend.

2. MM section needs to be rewritten:

a. What is n=9, n=3 so on? Elaborate.

b. Clean up agents: GCB in dspe tube as mentioned in materials/reagents does not match in this section.

c. A proper description on HLB cartridge should be given. How were they preconditioned? What was the suction? Which solvent was used to elute etc. Please see any previous paper.

d. In sample preparation, both cartridge and alumina were used? Nothing is mentioned in MM. Should be mentioned here. Why it was chosen? with whom it was compared?

e. Before injection to GCMS, the centrifuged material should have been filtered.

f. Please clarify how are going to get a concentration of 2.5 ng/ml in extract at 10 ppb fortification.

g. Why you have not used any water to saturate the sample? There are several advantages of using water. In addition, the recovery of extract (ACN/EA) will be very less in dry sample.

h. Nothing is mentioned on Matrix of three medicinal plants in MM section. whereas, most of the RD section is comparison of the three matrices.

i. Any statistics

3. Result:

a. At different time segments, pulsed injection pressure had different results? Why? No discussion is made.

b. Using EA, authors invited co-extracts as pointed out by the reviewers. Using water for saturation, this could have been avoided.

Reviewers' comments:

Reviewer's Responses to Questions

**Comments to the Author**

1. If the authors have adequately addressed your comments raised in a previous round of review and you feel that this manuscript is now acceptable for publication, you may indicate that here to bypass the “Comments to the Author” section, enter your conflict of interest statement in the “Confidential to Editor” section, and submit your "Accept" recommendation.

Reviewer #1: All comments have been addressed

Reviewer #3: All comments have been addressed

2. Is the manuscript technically sound, and do the data support the conclusions?

Reviewer #1: Yes

Reviewer #3: Partly

3. Has the statistical analysis been performed appropriately and rigorously? 

Reviewer #1: N/A

Reviewer #3: Yes

4. Have the authors made all data underlying the findings in their manuscript fully available?

Reviewer #1: Yes

Reviewer #3: Yes

5. Is the manuscript presented in an intelligible fashion and written in standard English?

Reviewer #1: Yes

Reviewer #3: Yes

6. Review Comments to the Author

Reviewer #1: (No Response)

Reviewer #3: 1. Authors have specified LOQ as low as 0.002 ppm for most of the pesticides analysed. But the recovery tested were well above the respective LOQ values, which is a mandatory requirement for method validation. Authors have quantified the pesticides at above LOQ level but the recovery at LOQ level is lacking. The supporting information in form of tested results may be supplied for more accuracy of the method.

2. The choice of solvents for extraction in a ratio has been already addressed by previous reviewers. The authors have tried to answer, but I have a doubt about the solubility of the pesticides in the mixture. Can the authors discuss about this.

3. As regards to the clean up steps, the authors have utilised alumina to remove the co-extractives. But, the authors have addressed the reviewers question, stating that ethyl acetate results in more extraction of co-extractives, as observed in increased dry weight. This creates ambiguity. The authors have first increased the co-extractives through extraction and then they are trying to remove the co-extractives using alumina. This requires justification.

4. TIC of control samples are presented but they should have also presented the TIC of fortified matrix.

7. PLOS authors have the option to publish the peer review history of their article (what does this mean?). If published, this will include your full peer review and any attached files.

Reviewer #1: **Yes: **Hyun Ho Noh

Reviewer #3: **Yes: **ABHIJIT KAR

---

## [Author Response · Author response to Decision Letter 1]

6 Jun 2023

<Author’s Response to Reviewer Comments for “PONE-D-22-33518R1”>

Journal Requirements:

Answer: Thank you for your advice regarding the references. Upon review, all references, except for reference number 6, are correct and up-to-date. For reference number 6, the original URL was expired, so we replaced it with the current, valid URL. We have carefully ensured no retracted papers were cited without proper indication. The detailed changes are noted in the attached rebuttal letter with our revised manuscript.

Academic Editor:

1. Thorough English editing is needed. Difficult to follow the write up. For example, see the line number 29-30 in abstract. I am unable to comprehend.

Answer: Thank you for your valuable comments. Our draft manuscript was proofread by Editage, a renowned English editing service. However, we recognized that there were still some awkward sentences. Therefore, we have entrusted our manuscript to an English language expert and have undertaken a thorough re-editing process, including line 29-30, to improve the overall clarity of the text.

2. MM section needs to be rewritten:

a. What is n=9, n=3 so on? Elaborate.

b. Clean up agents: GCB in dspe tube as mentioned in materials/reagents does not match in this section.

c. A proper description on HLB cartridge should be given. How were they preconditioned? What was the suction? Which solvent was used to elute etc. Please see any previous paper.

d. In sample preparation, both cartridge and alumina were used? Nothing is mentioned in MM. Should be mentioned here. Why it was chosen? with whom it was compared?

e. Before injection to GCMS, the centrifuged material should have been filtered.

f. Please clarify how are going to get a concentration of 2.5 ng/ml in extract at 10 ppb fortification.

g. Why you have not used any water to saturate the sample? There are several advantages of using water. In addition, the recovery of extract (ACN/EA) will be very less in dry sample.

h. Nothing is mentioned on Matrix of three medicinal plants in MM section. whereas, most of the RD section is comparison of the three matrices.

i. Any statistics

Answer: Thank you for your advice regarding the MM section.

a. The "n = 9" and "n = 3" in the text represent the number of repetitions for each experiment. To clarify this point, I have revised the paragraph in Lines 163-164 (This process was repeated nine times (n = 9) for each type of solvent.) and Lines 169-170 (This study was also repeated three times (n = 3) for each type of samples.).

b. Thank you for your careful reading and for pointing out the discrepancy related to the amount of GCB in d-SPE tubes mentioned in different sections of the manuscript. Upon reviewing, we found that the correct amounts of GCB are as stated in the first section of the manuscript, which are 2.5 mg and 7.5 mg for tubes 5982-5221 and 5982-5321 respectively. Therefore, the respective sentences in the second section in Lines (180-181) "The extracted samples from the optimized extraction step were purified using various types of d-SPE sorbents... (5) 25 mg PSA and 25 mg GCB, (6) 50 mg PSA and 50 mg GCB..." should be corrected as follows: "(5) 25 mg PSA and 2.5 mg GCB, (6) 25 mg PSA and 7.5 mg GCB."

c. In response to your query about the HLB cartridge, we used the Oasis PRiME HLB cartridge “Plus Light” in our study. This is a filter-type cartridge, which differs from the traditional cylindrical-type Oasis PRiME HLB. It specifically designed to simplify sample pretreatment. It does not require the traditional preconditioning, and thus the specific details about washing, suction, and elution solvent that you have asked for do not apply in this case. This feature is one of the primary reasons we selected this particular cartridge, as it streamlines the preparation process while maintaining a high degree of accuracy and precision. We have clearly specified "Oasis PRiME HLB Plus Light" in the manuscript to distinguish it from the traditional Oasis PRiME HLB. Zhang et al. (2018) have used HLB without any precondition, and we have described in Lines 198-199 that “According to the manufacturer's instructions and methods outlined in a previous paper [25], the organic supernatant (2 mL) was loaded into a syringe connected to the Oasis PRiME HLB plus light and passed through the cartridge.

d. I acknowledge the lack of detail in the materials and methods (MM) section concerning the use of cartridge and alumina in sample preparation. We described dual purification methods in Lines 186-189, that “In addition, a dual purification was conducted on the extract obtained from the No. (8) procedure by further implementing the No. (2) or (7) purification methods. The purification efficiency of each preparation method was compared in three types of herbal medicines, taking into account both recovery rates and matrix effects”. To increase the removal efficiencies of across a wide ranges of interference sample matrices, dual purifications were considered. This strategy enables the elimination of impurities that a single purification method cannot remove, by employing an additional purification precedure. Gong et al. (2020) demonstrated that the combining dSPE and Oasis PRiME HLB resulted in superior matrix removal, compared to using HLB alone. In this study, we implemented a dual purification strategy by conducting an Oasis PRiME HLB plus light cleanup, followed by d-SPE containing C18 or Alumina sorbents. We described the statements in Lines 335-340.

e. Our skilled researchers took precautions to avoid introducing solid impurities when transferring extracts for GC-MS/MS analysis from dSPE, so filtration step was not required. In multiresidue study, some of pesticides may be caught by filter.

f. 10 ug pesticide/kg (ppb) in 5 g sample was extracted in 10 mL of extraction solvent, so the concentration became 5 ug pesticide/L extract solution. Without concentration or dilution of solution, the extract was subjected to matrix-matched (extract/solvent, 1:1, v/v), so concentration of pesticide in the final extract solution became 2.5 ug/L extraction solution. We added the following sentence to Lines 204-205 to clarify the correlation between the sample and the final extract: "The sample was equivalent to 0.25 g per 1 mL in the final extract."

g. Before extracting pesticides from the samples using ACN/EA, we thoroughly saturated the samples with distilled water to ensure that the pesticides within the sample particles were adequately extracted. To clarify the expression, we have modified "added" in Line 191 to "saturated.": The homogenized sample (5 g) in a 50-mL centrifuge tube was added saturated with 10 mL of distilled water for 30 min to ensure sufficient soaking. In addition, the water residue was isolated using liquid-liquid partitioning to capture polar compounds in aqueous layer. This crucial detail was not explicitly stated in our original manuscript, and we understand that this might have led to some confusion. We have now included this information in Lines 196-198.

h. According to the editor’s valuable comments, we have added the following sentences in Lines 174-176 (In the overall extraction studies, we compared the extraction patterns across three types of herbal medicines: C. officinale, R. glutinosa, and P. lactiflora) and Lines 187-189 (The purification efficiency of each preparation method was compared in three types of herbal medicines, taking into account both recovery rates and matrix effects)

i. In response to your inquiry about statistical methods, we used mean values and relative standard deviation in the interpretation of our research findings, without the use of advanced statistics. Consequently, specific mentions of statistical processing were not included.

3. Result:

a. At different time segments, pulsed injection pressure had different results? Why? No discussion is made.

b. Using EA, authors invited co-extracts as pointed out by the reviewers. Using water for saturation, this could have been avoided.

Answer: Thank you for your valuable comments regarding the Results section.

a. We understand the reviewer's concern about the different results at different time segments of pulsed injection pressure, which was not extensively discussed in our manuscript. Our team attempted to interpret the varying changes depending on the retention time. However, as we are not experts in the field of studying the mechanical movements of GC equipment, a clear interpretation of the flow dynamics and mechanical interactions is challenging. It is also notable that we couldn't find any relevant mention in other scientific papers that studied pulsed injection pressure. We acknowledge this as a limitation of our current study, and we are planning to conduct a follow-up research to elucidate this issue.

b. We used water in sufficient amounts to saturate the sample before extracting pesticides with organic solvents. The water used in this process was subsequently separated into water and organic solvent layers by performing liquid-liquid partitioning with QuEChERS salts. This procedure allowed us to confine polar co-extracts, such as sugars, in the water. This crucial detail was not explicitly stated in our original manuscript, and we understand that this might have led to some confusion. We have now included this information in Lines 196-199.

Reviewer #3:

1. Authors have specified LOQ as low as 0.002 ppm for most of the pesticides analysed. But the recovery tested were well above the respective LOQ values, which is a mandatory requirement for method validation. Authors have quantified the pesticides at above LOQ level but the recovery at LOQ level is lacking. The supporting information in form of tested results may be supplied for more accuracy of the method.

Answer: We greatly appreciate the reviewer's meticulous attention to our manuscript. we would like to clarify that the definition of LOQ is the minimum concentration at which a signal-to-noise ratio of 10 or more is satisfied in the chromatogram (Lines 209-210), and recovery tests are not separately required at this level. The LOQs of target pesticides in our study is a suggestion of the highest sensitivity that can be output in the developed analytical method. As the purpose of pesticide multiresidue analysis is to check rapidly whether non-registered pesticides exceed 0.01 mg/kg in crops. Therefore, it is crucial to have quantitation at this concentration. Therefore, it is typical in multiresidue analysis to perform recovery study with 0.01 mg/kg as the minimum concentration.

2. The choice of solvents for extraction in a ratio has been already addressed by previous reviewers. The authors have tried to answer, but I have a doubt about the solubility of the pesticides in the mixture. Can the authors discuss about this.

Answer: We are grateful for the reviewer's thoughtful question. The choice of solvents, acetonitrile (ACN) and ethyl acetate (EA), was not arbitrary. Both ACN and EA are organic solvents with high solubility for the pesticides. The solvents were selected based on their polarity and ability to extract a wide range of pesticide residues with varying polarities. In our preliminary tests, we confirmed that these solvents could efficiently solubilize all the pesticides we tested, leading to satisfactory recovery rates. We have included this additional detail in Lines (243-245) to provide a more comprehensive explanation of our solvent choice and ensure the scientific rigor of our method is accurately portrayed.

3. As regards to the clean up steps, the authors have utilised alumina to remove the co-extractives. But, the authors have addressed the reviewers question, stating that ethyl acetate results in more extraction of co-extractives, as observed in increased dry weight. This creates ambiguity. The authors have first increased the co-extractives through extraction and then they are trying to remove the co-extractives using alumina. This requires justification.

Answer: Thank you for your valuable comments. We understand the seeming contradiction the reviewer points out in terms of increasing co-extractives through extraction, and then trying to remove them using alumina. However, this process is essential to our approach and allows us to optimize multiresidual pesticide recoveries. In our methodology, we added ethyl acetate to the acetonitrile (ACN) extraction solvent, which we found to improve the recovery of certain pesticides that were not adequately extracted with ACN alone. While beneficial for pesticide recovery, indeed increased the extraction of co-extractives. To counteract this, we employed a dual purification process, effectively minimizing pesticide loss and maximizing purification efficiency. We understand that this step may seem paradoxical, but it enables a high recovery of pesticides that could not be achieved with the sole extraction by ACN. We sought to resolve any misunderstandings by explaining the principles of the established analytical method in Lines 351-356 before describing the validation of the method.

4. TIC of control samples are presented but they should have also presented the TIC of fortified matrix.

Answer: We understand the reviewer's suggestion to present the Total Ion Chromatogram (TIC) of the fortified matrix. However, the fortified matrix includes peaks from all 296 target pesticides, which makes it extremely difficult to distinguish individual peaks in one TIC. We have confirmed that the peaks of the 296 pesticides are distinctly distinguishable from the baseline in the three herbal matrices.

---

## [Editor Report · Decision Letter 2]

21 Jun 2023

Simultaneous analytical method for 296 pesticide multiresidues in root and rhizome based herbal medicines with GC-MS/MS

PONE-D-22-33518R2

Dear Dr. Choi,

We’re pleased to inform you that your manuscript has been judged scientifically suitable for publication and will be formally accepted for publication once it meets all outstanding technical requirements.

Kind regards,

Totan Adak

Academic Editor

PLOS ONE
---

## [Editor Report · Acceptance letter]

26 Jun 2023

PONE-D-22-33518R2 

Simultaneous analytical method for 296 pesticide multiresidues in root and rhizome based herbal medicines with GC-MS/MS 

Dear Dr. Choi:

I'm pleased to inform you that your manuscript has been deemed suitable for publication in PLOS ONE. Congratulations! Your manuscript is now with our production department. 

Kind regards, 

on behalf of

Dr. Totan Adak 

Academic Editor

PLOS ONE